# Verifiably Forgotten? Gradient Differences Still Enable Data Reconstruction in Federated Unlearning

## Abstract

Federated Unlearning (FU) has emerged as a critical compliance mechanism for data privacy regulations, requiring unlearned clients to provide verifiable Proof of Federated Unlearning (PoFU) to auditors upon data removal requests. However, we uncover a significant privacy vulnerability: when gradient differences are served as PoFU, *honest-but-curious* auditors may exploit mathematical correlations between gradient differences and forgotten samples to reconstruct the latter. Such reconstruction, if feasible, would face three key challenges: (i) restricted auditor access to client-side data, (ii) limited samples derivable from individual PoFU, and (iii) high-dimensional redundancy in gradient differences. To overcome these challenges, we propose **I**nverting **G**radient difference to **F**orgotten data (IGF), a novel learning-based reconstruction attack framework that employs Singular Value Decomposition (SVD) for dimensionality reduction and feature extraction. IGF incorporates a tailored pixel-level inversion model optimized via a composite loss that captures both structural and semantic cues. This enables efficient and high-fidelity reconstruction of large-scale samples, surpassing existing methods. To counter this novel attack, we design an orthogonal obfuscation defense that preserves PoFU verification utility while preventing sensitive forgotten data reconstruction. Experiments across multiple datasets validate the effectiveness of the attack and the robustness of the defense. The code is available at https://anonymous.4open.science/r/IGF.

## 1 Introduction

Federated Learning (FL) enables distributed entities, such as financial institutions, healthcare providers, and IoT networks, to collaboratively train models without sharing raw data. This decentralized approach mitigates risks associated with data transfer, enhancing privacy and security for data owners. However, regulations like the GDPR (Rosen, 2011; Pardau, 2018), which enshrine the *right to be forgotten*, pose a significant technical challenge for FL systems. Merely preventing raw data leaks is insufficient for compliance. Instead, the requirement to honor data subjects' requests for erasure (Article 17, GDPR) necessitates mechanisms to eliminate impacts resulting from specific personal data on the global models and demonstrate effective erasure. This challenge has spurred the development

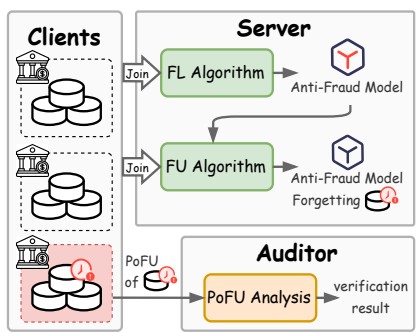

Figure 1: Auditing process in FU

of verifiable Federated Unlearning (FU) (Liu et al., 2020), a paradigm designed to verifiably forget the contribution of designated data from trained models.

Figure 1 illustrates a typical scenario where multinational financial institutions, acting as FL clients 🏦, collaboratively train an anti-fraud model (Lindstrom, 2024). Subsequently, the auditor mandates all clients to forget the outdated transaction data 🗑️ using the FU algorithm and obtains proof of FU (PoFU) (Gao et al., 2024; Weng et al., 2024; Zuo et al., 2025; Salem et al., 2020) from the unlearned client. Given that auditors lack direct access to raw client data, they typically rely on PoFU as a

non-invasive auditing mechanism. In particular, PoFU often leverages gradient differences, defined as the gradient of the forgotten sample (e.g., outdated transactions here) computed on the original model minus that on the unlearned model. A sufficiently small L2 norm of this difference indicates successful forgetting (Gao et al., 2024).

However, most research (Wu et al., 2022; Wang et al., 2022; Zhong et al., 2025) primarily focuses on FU algorithm design, overlooking vulnerabilities to reconstruction attacks by third-party auditors (Boenisch et al., 2023; Le et al., 2023), especially when gradient differences serve as PoFU. Recent advances in reconstruction attacks have exposed critical vulnerabilities in centralized machine (un)learning. For instance, DLG (Zhu et al., 2019) showed that shared gradients can be inverted to reconstruct training data, while subsequent work (Geiping et al., 2020) highlighted privacy leakage risks from gradient sharing. More recently, unlearning inversion attacks (Hu et al., 2024) reconstruct forgotten data by only accessing the parameter deviations of the original and unlearned models. However, these approaches face three primary limitations when applying to FU scenarios: (i) they require *white-box access* to both models to compute parameter deviations, (ii) they struggle with *large-scale* data reconstruction due to limited, noisy gradient differences weakly linked to forgotten samples, requiring novel inversion methods, and (iii) the *high dimensionality* of parameter deviations or gradients differences increases the computational cost of inversion models. More crucially, as the auditor lacks access to client-side raw data (Thudi et al., 2022) and relies solely on PoFU to audit unlearning, there are additional complexities to be considered for reconstruction attack. Current reconstruction attacks target model parameters or gradients, but those exploiting gradient differences that are commonly used for PoFU remain underexplored. This gap motivates our research question:

*Q: Can gradient differences, serving as PoFU, enable third-party auditors to reconstruct forgotten data? If so, how can high-fidelity, large-scale data reconstruction be achieved against high-dimensional gradient differences?*

To address this, we propose a learning-based reconstruction attack for verifiable FU, termed **I**nverting **G**radient difference to **F**orgotten data (IGF). To handle high-dimensional gradient differences, we employ Singular Value Decomposition (SVD) for dimensionality reduction, extracting essential features while eliminating redundancy, thus streamlining the input of the inversion model. We then design a pixel-level convolutional inversion model that learns the latent mapping between gradient differences and original samples, optimized via a composite loss function that balances structural and perceptual fidelity. This enables batch-wise reconstruction from individual PoFU, avoiding per-sample optimization overhead. Collectively, these components facilitate robust, large-scale reconstruction across benchmark datasets and global model architectures. Our contributions include:

- We identify gradient differences that serve as PoFU for a novel attack surface capable of high-fidelity data reconstruction. By formalizing an *honest-but-curious* third-party auditor, we demonstrate that passive observers can reconstruct forgotten samples during the critical FU auditing phase (Li et al., 2022).

- We propose the IGF attack framework, integrating SVD with a pixel-level inversion network optimized via a composite loss function. Extensive experiments and ablation studies show that IGF outperforms state-of-the-art (SOTA) learning-based (LTI (Wu et al., 2023)) and optimization-based (GIAMU (Hu et al., 2024), DLGD (Zhu et al., 2019)) methods, achieving superior reconstruction fidelity and computational efficiency.

- We further propose an orthogonal obfuscation defense mechanism to mitigate IGF and validate defense efficacy through rigorous theoretical analysis and comprehensive experiments.

## 2 RELATED WORK

**Federated Unlearning (FU).** FU has recently emerged to address the challenge of selectively removing specific clients or data points from a trained FL model. This paradigm is driven by regulatory imperatives, such as the *right to be forgotten* under GDPR, as well as the inherent dynamism of real-world FL deployments. Existing approaches can be categorized into two main types: *Exact Federated Unlearning (EFU)* (Liu et al., 2022) and *Approximate Federated Unlearning (AFU)* (Halimi et al., 2022). EFU achieves thorough removal by retraining the model from scratch on the retained dataset, ensuring that the influence of the target data is completely eliminated. However, this method is computationally intensive and may be impractical for large-scale FL systems. AFU

aims to reduce computational overhead by approximating the unlearning process through applying gradient ascent to maximize the loss. For instance, Wang et al. (2024) propose that clients estimate the gradient influence of the data to be removed using local retained data and then apply gradient ascent to negate this influence. A subsequent fine-tuning step is introduced to preserve overall utility. Similarly, Xu et al. (2024) employ model explanations to identify key parameter channels associated with the forgotten categories and update only those channels in reverse. Meanwhile, Gu et al. (2024) pre-generate linear transformation parameters related to the target data during the training phase and applies reverse transformations to eliminate unwanted effects. The above methods balance effectiveness and efficiency. Some studies (Chen et al., 2025; Wang et al., 2025) explore how to diminish the model's utility by poisoning or cause excessive forgetting through malicious requests, yet overlook potential reconstruction vulnerabilities during the verification stage.

**Gradient Inversion Attack.** Recent studies have leveraged gradient inversion techniques to recon-struct clients' private training data in FL (Zhang et al., 2023; Jeon et al., 2021; Fang et al., 2023; Sun et al., 2024; Wu et al., 2023). Zhang et al. (2023) demonstrate the feasibility of generative gradient inversion in FL by constructing an over-parameterized convolutional neural network that satisfies gradient-matching requirements. Similarly, Jeon et al. (2021) leverage pre-trained generative models as priors to circumvent direct optimization in high-dimensional pixel space and reconstructs data via latent-space parameter optimization. Additionally, Fang et al. (2023) adopt a staged optimization strategy for the intermediate feature domains of generative models, progressively optimizing from the latent space to intermediate layers to enhance attack effectiveness. Sun et al. (2024) introduce an anomaly detection model to capture latent distributions from limited data, using it as a regularization term to improve attack performance. In the context of FU, Hu et al. (2024) reveal the feature and label information by analyzing differences between the original and unlearned models.

Therefore, traditional gradient inversion attacks focus on reconstructing training data directly from original gradients provided by clients in standard FL scenarios. In contrast, our work targets **gradient differences** used as PoFU, where the attacker must reconstruct deleted data from indirect and variant gradient information. *This introduces unique challenges: gradient differences contain limited and mixed signals with weaker correlations to the forgotten samples, requiring fundamentally different inversion methods.*

## 3 METHODOLOGY

### 3.1 PROBLEM FORMULATION

**Federated Learning (FL).** In the FL framework with $H$ clients, each client $i$ ($i \in [H]$) holds a local dataset $\mathcal{D}_i$ containing $|\mathcal{D}_i|$ samples. Let $\mathbf{M}$ denote the original global model parameterized by $\boldsymbol{\theta}$, and consider a supervised learning objective that minimizes the empirical loss over the federated dataset $\mathcal{D} = \bigcup_{i=1}^{H} \mathcal{D}_i$: $\mathcal{L}(\boldsymbol{\theta}) = \frac{1}{|\mathcal{D}|} \sum_{(x,y) \in \mathcal{D}} \ell\big(\mathbf{M}(x; \boldsymbol{\theta}), y\big)$. The stochastic gradient for a data sample $(x_s, y_s) \in \mathcal{D}$ is $\mathbf{g}_s = \nabla_{\boldsymbol{\theta}} \ell\big(\mathbf{M}(x_s; \boldsymbol{\theta}), y_s\big)$. Federated Averaging (FedAvg) (McMahan et al., 2017) operates through $T$ global rounds. At each global round $t \in \{0, 1, \ldots, T-1\}$, the server broadcasts the current global model parameters $\boldsymbol{\theta}^t$ to all clients. Each client $i$ updates $\boldsymbol{\theta}^t$ via local SGD on $\mathcal{D}_i$: $\boldsymbol{\theta}_i^t = \boldsymbol{\theta}^t - \eta \cdot \nabla_{\boldsymbol{\theta}} \mathcal{L}_i(\boldsymbol{\theta}^t)$, where $\mathcal{L}_i(\boldsymbol{\theta}^t) = \frac{1}{|\mathcal{D}_i|} \sum_{(x,y) \in \mathcal{D}_i} \ell\big(\mathbf{M}(x; \boldsymbol{\theta}^t), y\big)$. Server aggregates via weighted averaging:

$$\boldsymbol{\theta}^{t+1} = \sum_{i=1}^{H} \frac{|\mathcal{D}_i|}{|\mathcal{D}|} \boldsymbol{\theta}_i^t, \quad |\mathcal{D}| = \sum_{i=1}^{H} |\mathcal{D}_i|. \tag{1}$$

The final global model parameter after $T$ rounds is $\boldsymbol{\theta}^T$.

**FU Scenarios.** Let $\mathcal{C}_n \subseteq [H]$ denote normal clients retaining their original datasets $\{\mathcal{D}_j\}_{j \in \mathcal{C}_n}$, and $\mathcal{C}_u = [H] \setminus \mathcal{C}_n$ represent unlearned clients modifying their local datasets $\{\mathcal{D}_i\}_{i \in \mathcal{C}_u}$. Following Zhong et al. (2025), we formalize three scenarios: (i) *sample-level unlearning:* For each client $i \in \mathcal{C}_u$, partition $\mathcal{D}_i$ into retained $\mathcal{D}_i^r$ and forgotten subsets $\mathcal{D}_i^f = \mathcal{D}_i \setminus \mathcal{D}_i^r$; (ii) *class-level unlearning:* Each client $i \in \mathcal{C}_u$ removes all samples of target class $y^f$, yielding $\mathcal{D}_i^f = \{(x,y) \in \mathcal{D}_i \mid y = y^f\}$ with $\mathcal{D}_i^r = \mathcal{D}_i \setminus \mathcal{D}_i^f$; (iii) *client-level unlearning:* Each client $i \in \mathcal{C}_u$ sets $\mathcal{D}_i^f = \mathcal{D}_i$ and $\mathcal{D}_i^r = \emptyset$. We denote the unlearned global model as $^u\mathbf{M}$, the forgotten dataset as $\mathcal{D}^{\text{forgotten}} = \bigcup_{i \in \mathcal{C}_u} \mathcal{D}_i^f$ and the retained dataset as $\mathcal{D}^{\text{retained}} = \big(\bigcup_{j \in \mathcal{C}_n} \mathcal{D}_j\big) \cup \big(\bigcup_{i \in \mathcal{C}_u} \mathcal{D}_i^r\big)$.

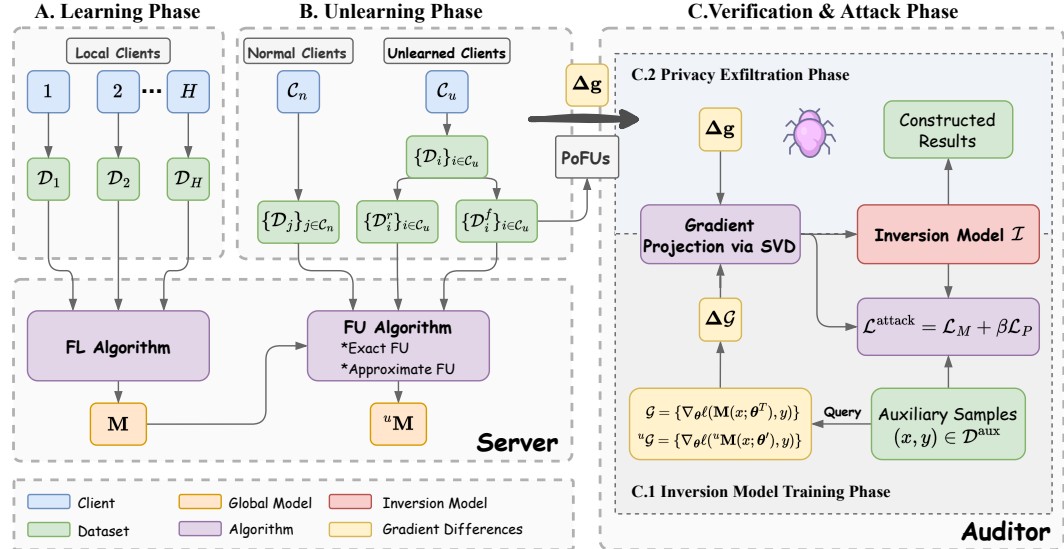

Figure 2: Schematic overview of IGF framework. **A. Learning Phase**: Clients collaboratively train the global model via FL. **B. Unlearning Phase**: The unlearned clients are required to forget specific data contributions and submit the proof of federated unlearning (PoFU). **C. Verification & Attack Phase**: The *honest-but-curious* auditor $\mathcal{A}$ verifies PoFUs, while attempting to infer forgotten data using a pre-trained inversion model $\mathcal{I}$.

**FU Methods.** We implement two mainstream FU approaches: (i) **EFU** retrains the global model on dataset $\mathcal{D}^{\text{retained}}$ from scratch, minimizing $\sum_{(x,y)\in\mathcal{D}^{\text{retained}}} \ell(\mathbf{M}(x;\boldsymbol{\theta}), y)$. This method precisely removes contributions of $\mathcal{D}^{\text{forgotten}}$ from the global model. (ii) **AFU** performs projected gradient ascent and constrains maximization on $\mathcal{D}^{\text{forgotten}}$. For each client $i \in \mathcal{C}_u$, it computes $\boldsymbol{\theta}'_i = \boldsymbol{\theta}^T + \eta_u \cdot \nabla_{\boldsymbol{\theta}} \mathcal{L}'_i(\boldsymbol{\theta}^T)$ where $\mathcal{L}'_i(\boldsymbol{\theta}^T) = \frac{1}{|\mathcal{D}_i^f|} \sum_{(x,y)\in\mathcal{D}_i^f} \ell({}^u\mathbf{M}(x;\boldsymbol{\theta}^T), y)$ but maintains $\|\boldsymbol{\theta}'_i - \boldsymbol{\theta}^T\|_2 \leq \zeta$, where $\zeta$ is the parameter deviation constraint. Then the server aggregates the unlearned local model parameters:

$$\boldsymbol{\theta}' = \sum_{i \in \mathcal{C}_u} \frac{|\mathcal{D}_i^f|}{|\mathcal{D}^{\text{forgotten}}|} \boldsymbol{\theta}'_i, \quad |\mathcal{D}^{\text{forgotten}}| = \sum_{i \in \mathcal{C}_u} |\mathcal{D}_i^f|, \tag{2}$$

and fine-tunes ${}^u\mathbf{M}$ with $\boldsymbol{\theta}'$ on $\mathcal{D}^{\text{retained}}$.

**Verification in FU.** Each unlearned client $i \in \mathcal{C}_u$ locally computes PoFU of gradient differences $\Delta\mathbf{g}^{(n_i)} = \left\{ \Delta\mathbf{g}_j^{(n_i)} = \nabla_{\boldsymbol{\theta}} \ell(\mathbf{M}(x_j;\boldsymbol{\theta}^T), y_j) - \nabla_{\boldsymbol{\theta}} \ell({}^u\mathbf{M}(x_j;\boldsymbol{\theta}'), y_j) | (x_j, y_j) \in \mathcal{D}_i^f \right\}$. Auditor receives PoFUs $\Delta\mathbf{g} = \{\Delta\mathbf{g}^{(n_i)}\}_{i \in \mathcal{C}_u}$ and validates unlearning by checking each $\|\Delta\mathbf{g}_j^{(n_i)}\|_2 \leq \tau$ with predefined threshold $\tau$ (Gao et al., 2024). The necessity of the gradient differences in verifiable FU lies in ensuring that a data point $(x, y)$ is included in the training dataset of the original model $\mathbf{M}$ but excluded from that of the unlearned model ${}^u\mathbf{M}$.

**Threat Assumption.** We model the auditor, denoted $\mathcal{A}$, as an *honest-but-curious* entity that strictly follows the FU protocol but seeks to infer private client data. Consistent with prior reconstruction attacks (Wu et al., 2023; Hu et al., 2024; Geiping et al., 2020; Zhu et al., 2019), $\mathcal{A}$ possesses an auxiliary dataset $\mathcal{D}^{\text{aux}}$. Operating in a gray-box setting, $\mathcal{A}$ lacks knowledge of the global model's architecture but can collude with the server to query the flattened gradient for arbitrary samples from both the original model $\mathbf{M}$, and the unlearned model ${}^u\mathbf{M}$. During the exploitation phase, $\mathcal{A}$ passively collects PoFUs $\Delta\mathbf{g}$ from unlearned clients, and endeavors to reconstruct the forgotten samples.

## 3.2 FRAMEWORK OF IGF

We adopt a learning-based inversion model to invert gradient differences to forgotten samples during the verification phase of FU. The main schematic of IGF is shown in Figure 2, and the formalized details are as follows:

**Inversion Model Training Phase.** (i) **Preparation of Training Dataset**. To prepare the training data for inversion model $\mathcal{I}$, for each data point $(x_i, y_i)$ in auxiliary dataset $\mathcal{D}^{\text{aux}}$, the auditor $\mathcal{A}$ collects:

$$
\begin{cases}
\mathcal{G}_i = \{\nabla_{\boldsymbol{\theta}}\ell(\mathbf{M}(x;\boldsymbol{\theta}^T), y_i)\}_{(x_i,y_i)\in\mathcal{D}^{\text{aux}}} \\
{}^u\mathcal{G}_i = \{\nabla_{\boldsymbol{\theta}}\ell({}^u\mathbf{M}(x;\boldsymbol{\theta}'), y_i)\}_{(x_i,y_i)\in\mathcal{D}^{\text{aux}}},
\end{cases}
\tag{3}
$$

where $\mathcal{G}_i$ and ${}^u\mathcal{G}_i$ denote the sets of flatten gradients queried from $\mathbf{M}$ and ${}^u\mathbf{M}$, respectively. Gradient differences $\boldsymbol{\Delta}\mathcal{G} = \{\boldsymbol{\Delta}\mathcal{G}_i = \mathcal{G}_i - {}^u\mathcal{G}_i | (x_i, y_i) \in \mathcal{D}^{\text{aux}}\}$ form a set of $d$-dimensional vectors, with $d$ as the number of trainable parameters.

(ii) **Gradient Differences Projection via SVD**. To extract the key features and address redundancy caused by the high dimensionality of gradient differences, $\mathcal{A}$ projects $\boldsymbol{\Delta}\mathcal{G}$ to a lower-dimensional space using SVD. Let the $m$ denote the number of samples in $\mathcal{D}^{\text{aux}}$, $\mathcal{A}$ constructs a matrix $\boldsymbol{\Psi} = [\boldsymbol{\Delta}\mathcal{G}_1, \boldsymbol{\Delta}\mathcal{G}_2, \ldots, \boldsymbol{\Delta}\mathcal{G}_m]^\top \in \mathbb{R}^{m\times d}$, where each row corresponds to a sample's gradient difference and $m \ll d$ typically holds. $\mathcal{A}$ centers the gradient differences by subtracting the mean vector $\boldsymbol{\mu} = \frac{1}{m}\sum_{i=1}^m \boldsymbol{\Delta}\mathcal{G}_i$, resulting in $\boldsymbol{\Psi}^{\text{cen}} = \boldsymbol{\Psi} - \boldsymbol{\mu}\mathbf{1}_m^\top$. Then $\mathcal{A}$ then performs SVD on $\boldsymbol{\Psi}^{\text{cen}}$, yielding $\boldsymbol{\Psi}^{\text{cen}} = \mathbf{U}\boldsymbol{\Sigma}\mathbf{V}^\top$ with $\mathbf{U} \in \mathbb{R}^{m\times m}$, $\mathbf{V} \in \mathbb{R}^{d\times d}$, and diagonal matrix $\boldsymbol{\Sigma}$ contains singular values $\sigma_1 \geq \sigma_2 \geq \cdots \geq \sigma_m \geq 0$. To preserve essential information while reducing dimensionality, $\mathcal{A}$ selects the smallest $k$ such that the cumulative explained variance exceeds a threshold $\nu$:

$$
k = \min\left\{ j \,\middle|\, \sum_{i=1}^j \sigma_i^2 / \sum_{i=1}^m \sigma_i^2 \geq \nu \right\}.
\tag{4}
$$

So $\mathcal{A}$ gets the projection matrix $\mathbf{V}^{[k]} \in \mathbb{R}^{d\times k}$ denotes the first $k$ columns of $\mathbf{V}$. And the projected gradient differences of $\mathcal{D}^{\text{aux}}$ are computed as $\boldsymbol{\Delta}\mathcal{G}^{\text{proj}} = \boldsymbol{\Psi}\mathbf{V}^{[k]} \in \mathbb{R}^{m\times k}$.

(iii) **Training Inversion Model**. $\mathcal{A}$ trains the inversion model, denoted as $\mathcal{I}$ and parameterized by $\boldsymbol{\omega}$, to map projected gradient differences to samples in $\mathcal{D}^{\text{aux}}$ by minimizing the composite loss function:

$$
\mathcal{L}^{\text{attack}}(\boldsymbol{\omega}) = \mathcal{L}_M(\boldsymbol{\omega}) + \beta\mathcal{L}_P(\boldsymbol{\omega}),
\tag{5}
$$

where $\beta$ trades off between pixel-level accuracy and perceptual quality. This design is common in image reconstruction tasks and can flexibly adjust the optimization objectives of the model to ensure that the reconstruction results are both accurate and natural. specifically, $\mathcal{L}_M$ quantifies the structural pixel-level discrepancy between reconstructed image $\mathcal{I}(\boldsymbol{\Delta}\mathcal{G}_i^{\text{proj}}; \boldsymbol{\omega})$ and ground truth image $x_i$:

$$
\mathcal{L}_M(\boldsymbol{\omega}) = \frac{1}{m}\sum_{i=1}^m \|\mathcal{I}(\boldsymbol{\Delta}\mathcal{G}_i^{\text{proj}}; \boldsymbol{\omega}) - x_i\|_2^2.
\tag{6}
$$

Similarly, we define $\mathcal{L}_P$, which measures the semantic similarity between the reconstructed and true images using a VGG-based feature extractor $\phi(\cdot)$:

$$
\mathcal{L}_P(\boldsymbol{\omega}) = \frac{1}{m}\sum_{i=1}^m \|\phi\left(\mathcal{I}(\boldsymbol{\Delta}\mathcal{G}_i^{\text{proj}}; \boldsymbol{\omega})\right) - \phi(x_i)\|_2^2
\tag{7}
$$

Further, we elaborately designed the architecture of $\mathcal{I}$ to capture the latent mapping between gradient differences and images effectively. $\mathcal{I}$ employs a pixel-level convolutional network for progressive upsampling, which reduces artifacts in the reconstructed images. This design facilitates a nonlinear transformation from PoFU space to structured image space. Further architectural details are provided in Appendix F.

**Privacy Exfiltration Phase.** Following the training phase, the auditor $\mathcal{A}$ possesses the projection matrix $\mathbf{V}^{[k]}$ and the inversion model $\mathcal{I}$ with parameter $\boldsymbol{\omega}$. Upon receiving PoFUs, for each PoFU $\boldsymbol{\Delta}\mathbf{g}^{(n_i)}$ of each client $i \in \mathcal{C}_u$, $\mathcal{A}$ constructs the matrix $\boldsymbol{\Psi}^{(n_i)} = \left[\boldsymbol{\Delta}\mathbf{g}_1^{(n_i)}, \boldsymbol{\Delta}\mathbf{g}_2^{(n_i)}, \ldots, \mathbf{g}_{n_i}^{(n_i)}\right]^\top \in \mathbb{R}^{n_i\times d}$, where $n_i$ denotes the number of samples in $\mathcal{D}_i^f$. This matrix is then projected into a lower-dimensional space $\boldsymbol{\Delta}\mathbf{g}^{(n_i)^{\text{proj}}} = \boldsymbol{\Psi}^{(n_i)}\mathbf{V}^{[k]} \in \mathbb{R}^{n_i\times k}$. The batched reconstruction of projected gradient differences $\boldsymbol{\Delta}\mathbf{g}^{(n_i)^{\text{proj}}}$ is performed as follows:

$$\hat{\mathbf{x}}^{(n_i)} = \{\hat{x}_j = \mathcal{I}(\mathbf{\Delta g}_j^{(n_i)\mathrm{proj}}; \boldsymbol{\omega}) | j \in [n_i]\}, \tag{8}$$

where $\hat{\mathbf{x}}^{(n_i)} = \{\hat{x}_1, \hat{x}_2, \ldots, \hat{x}_{n_i}\}$ represents the $n_i$ reconstructed samples of client $i$. This exploitation enables $\mathcal{A}$ to utilize the pre-trained inversion model to implement the large-scale reconstructions from individual PoFU, thereby compromising data privacy even from the passive view.

### 3.3 ORTHOGONAL OBFUSCATION DEFENSE METHOD

Our inversion model exploits the directional information in gradient differences to reconstruct sensitive training data. Traditional defense methods often fail to disrupt the directional patterns, preserving the overall gradient differences structure and remaining susceptible to statistical recovery techniques. As illustrated in Figure 3, we propose a defense strategy that alters the vector direction while retaining the L2-norm information necessary for auditing. Our approach projects gradient differences into an orthogonal subspace, thereby disrupting the patterns and spatial structures that attackers rely on to reconstruct the forgotten sample.

For each PoFU $\mathbf{\Delta g}^{(n_i)}$ of unlearned client $i$, $i$ needs to modify the direction of each entry $\mathbf{\Delta g}_j^{(n_i)}$ but maintain its L2-norm. We introduce random vectors $\mathbf{r}^{(n_i)}$ that are orthogonal to $\mathbf{\Delta g}^{(n_i)}$ element-wisely. The construction begins by sampling an initial random vector $\mathbf{r}_j^{(n_i)}$ with the same dimensionality as $\mathbf{\Delta g}_j^{(n_i)}$, drawn from a standard normal distribution $\mathbf{r}_j^{(n_i)} \sim \mathcal{N}(0,1)^d$. Then client $i$ applies the Gram-Schmidt orthogonalization (ort, 2001) to compute:

$$\mathbf{\Delta g}_j^{(n_i)\mathrm{obf}} = \mathbf{r}_j^{(n_i)} - \frac{\mathbf{r}_j^{(n_i)\top} \mathbf{\Delta g}_j^{(n_i)}}{\|\mathbf{\Delta g}_j^{(n_i)}\|^2} \mathbf{\Delta g}_j^{(n_i)}. \tag{9}$$

Figure 3: Schematic of orthogonal obfuscation defense

This step ensures that $\mathbf{\Delta g}_j^{(n_i)\mathrm{obf}}$ lies in a subspace orthogonal to $\mathbf{\Delta g}_j^{(n_i)}$, effectively decoupling its direction from the original PoFU vector while preserving the randomness needed for obfuscation.

## 4 EXPERIMENT

### 4.1 EXPERIMENT SETTINGS

**Datasets and Models.** We assess the IGF framework on widely adopted benchmark datasets: CIFAR-10, CIFAR-100 (Krizhevsky et al., 2009), MNIST (LeCun et al., 1998), and Fashion-MNIST (Xiao et al., 2017). These datasets offer diverse challenges, featuring varying image resolutions ($28 \times 28$ and $32 \times 32$) and class numbers (10 to 100), making them an ideal testbed for assessing generalization. Additionally, we utilize the SVHN dataset (Netzer et al., 2011) as the out-of-distribution (OOD) auxiliary dataset to evaluate the attack's robustness to distributional shifts. More dataset details are provided in Appendix C.1. To probe the attack's robustness across architectural variations and to explore how the proposed inversion model scales with the network complexity of the global model, we adopt two architectures: a convolutional neural network (*ConvNet*) and a deeper residual network (*ResNet20*) (He et al., 2016).

**Training Setup.** In cross-silo FL and FU, we configure 40 clients with $10\%$ client selection and conduct 20 global rounds to derive the original and unlearned models. For the unlearning task, we designate 1000 samples to be forgotten. We consider an *honest-but-curious* adversary $\mathcal{A}$ capable of storing or collecting a small auxiliary dataset, with a size comparable to a typical validation or test set, consistent with prior work (Sun et al., 2024; Wu et al., 2023). Furthermore, the auxiliary dataset is *in-distribution* with respect to the forgotten dataset.[1] During the attack, $\mathcal{A}$ trains the inversion model with $\beta = 1$, batch size 256, learning rate $10^{-4}$, and a fixed seed (1234) for reproducibility. Gradient

---

[1] Notably, high-quality reconstruction remains feasible even under out-of-distribution auxiliary data; refer to Appendix E.3 for ablation details.

differences from *ConvNet* are used directly, while those from *ResNet20* are compressed via SVD projection to reduce computation. All experiments are conducted in PyTorch on NVIDIA A10 GPUs. To assess IGF, we adopt standard reconstruction metrics: MSE, PSNR, and LPIPS (Geiping et al., 2020; Hu et al., 2024; Sun et al., 2024; Zhang et al., 2018); metric details appear in Appendix C.2.

## 4.2 EXPERIMENTAL RESULTS

Table 1: Reconstruction performance (MSE, PSNR, and LPIPS) on CIFAR-10 and CIFAR-100 datasets with *ConvNet* and *ResNet20* as global models. Gradient differences are applied with no defense. Each cell reports results for EFU / AFU, with **bold** indicating the best performance across different FU scenarios.

| Backbone | Method | FU Scenario | CIFAR-10 | | | CIFAR-100 | | |
| --- | --- | --- | --- | --- | --- | --- | --- | --- |
| | | | MSE ↓ | PSNR ↑ | LPIPS ↓ | MSE ↓ | PSNR ↑ | LPIPS ↓ |
| *ConvNet* | Ours | *sample-level* | 0.0211 / **0.0218** | 17.19 / **17.09** | **0.3261** / 0.3624 | **0.0364** / 0.0261 | **14.97** / 16.07 | 0.4383 / **0.4190** |
| | Ours | *class-level* | 0.0259 / 0.0234 | 16.08 / 16.51 | 0.3531 / 0.3316 | 0.0397 / 0.0298 | 14.41 / 15.73 | 0.4451 / 0.4201 |
| | Ours | *client-level* | **0.0206** / 0.0223 | **17.32** / 16.78 | 0.3747 / 0.3558 | 0.0382 / 0.0265 | 14.65 / **16.07** | **0.4361** / 0.4223 |
| | GIAMU | *sample-level* | 0.2330 / 0.2460 | 13.22 / 12.78 | 0.3390 / **0.3190** | – | – | – |
| *ResNet20* | Ours | *sample-level* | 0.0445 / 0.0564 | 14.05 / 13.02 | **0.4607** / 0.4719 | **0.0391** / 0.0353 | **14.56** / 15.02 | 0.4267 / 0.4025 |
| | Ours | *class-level* | 0.0535 / **0.0512** | 13.01 / **13.21** | 0.4608 / **0.4366** | 0.0474 / 0.0438 | 13.49 / 13.84 | **0.4060** / 0.4032 |
| | Ours | *client-level* | **0.0435** / 0.0533 | **14.12** / 13.08 | 0.4617 / 0.4983 | 0.0422 / 0.0362 | 14.27 / 14.73 | 0.4187 / **0.3627** |

**Reconstruction Performance across Datasets.** The results presented in Table 1 provide compelling evidence of IGF's capability to reconstruct forgotten data with high fidelity. On CIFAR-10 with *ConvNet* under EFU at the *sample-level*, IGF achieves an MSE of 0.0211, PSNR of 17.19, and LPIPS of 0.3261, reflecting reconstructions with minimal pixel-wise errors and superior perceptual quality. On the more complex CIFAR-100 dataset, which contains 100 fine-grained classes compared to CIFAR-10's 10, we observe a moderate decline in performance: MSE increases to 0.0364, PSNR decreases to 14.9658, and LPIPS rises to 0.4383. This performance degradation is consistent across all FU methods and scenarios with *ConvNet* on CIFAR-100, which we attribute to the increased dataset complexity and higher inter-class variability, making inversion inherently more challenging.

**Adaptability across Global Model Architectures.** IGF also demonstrates adaptability across model architectures. On CIFAR-10, reconstruction performances with *ConvNet* are slightly better than *ResNet20*, with MSE values of 0.0211 and 0.0445, respectively. Consistent with prior findings (Wu et al., 2023), inversion performance declines as FL model complexity increases. This disparity stems from the higher-dimensional gradients of *ResNet20*, which introduce greater noise and optimization challenges, thereby reducing inversion fidelity compared to the cleaner, more tractable gradients of *ConvNet*. Nevertheless, IGF achieves satisfactory reconstruction quality even with the deeper *ResNet20* architecture, highlighting its robustness to varying model complexities.

**Adaptability across FU Scenarios.** We test IGF under three FU scenarios: *sample-level unlearning* (the number of samples to be forgotten is set to 1000), *class-level unlearning* (the class index to be forgotten is set to 1), and *client-level unlearning* (all samples from the third client are set to be forgotten). IGF exhibits stable performance, with *ConvNet*'s MSE fluctuating within 0.0053 under EFU on CIFAR-10, indicating resilience to differing unlearning granularity. In other configurations, alterations to the FU scenarios have a negligible impact on reconstruction performance, further highlighting IGF's stability.

**Vulnerability Comparison of FU Methods.** Experimental results reveal certain gaps in vulnerability to reconstruction attacks between EFU and AFU methods. EFU outperforms AFU in reconstruction metrics on CIFAR-10 with *ResNet20*, as EFU's retraining from scratch yields clearer gradient differences reflecting forgotten data's impact. In contrast, AFU's gradient ascent operation introduces noise, complicating reconstruction. Despite this, IGF achieves reasonable reconstruction quality, highlighting a critical privacy risk: even AFU methods remain vulnerable to reconstruction attacks.

**Comparison with Baselines.** We first compare IGF against GIAMU (Hu et al., 2024), a recent inversion attack specifically tailored for centralized machine unlearning. GIAMU takes the difference between the original and unlearned models as input and reconstructs samples via an optimization-based approach. As shown in Table 1, all GIAMU results are directly sourced from Hu et al. (2024) where the training datasets of the two models differ by only a single sample. For *sample-level* unlearning on CIFAR-10 under EFU, IGF outperforms GIAMU by 88.1%, 30.1%, and 3.8% in MSE, PSNR, and LPIPS, respectively. Under AFU, the improvements are even more substantial, with gains

of 91.1% in MSE and 33.6% in LPIPS. Moreover, as with other optimization-based methods (Zhu et al., 2019; Ju et al., 2025), GIAMU requires hundreds of queries per sample during the online (Privacy Exfiltration) phase, whereas IGF needs only two queries. Additionally, GIAMU assumes a white-box setting in which $\mathcal{A}$ can access the parameters of the original and unlearned models, which significantly complicates the attack process.

Notably, few baseline attacks are suitable for direct comparison in FU reconstruction, as most methods target gradients derived from fully trained models, **we nonetheless present a comprehensive comparison of IGF with two SOTA inversion approaches: the learning-based LTI (Wu et al., 2023) and the optimization-based DLGD (Zhu et al., 2019), detailed in Appendix D**.

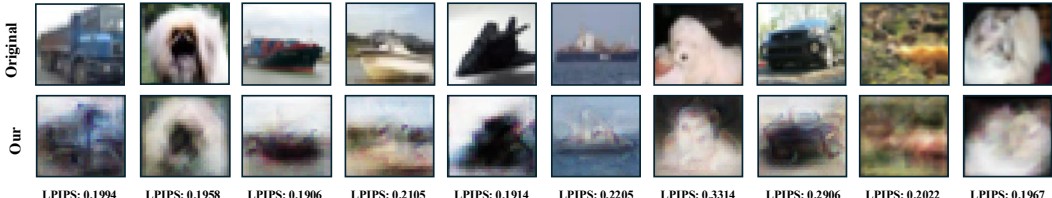

Figure 4: Original and reconstructed images from the CIFAR-10 dataset, with 1,000 forgotten samples at the *sample-level* using EFU.

**Visual Inspection of Reconstructed Images.** Beyond quantitative metrics, visual inspection of the reconstructed images in Figure 4 offers additional insights into IGF's effectiveness. The reconstructed images clearly capture the essential features of the original forgotten samples, including object shapes, colors, and textures. This visual similarity reinforces the quantitative results and shows that our attack can reconstruct forgotten data with sufficient fidelity to pose a real privacy risk. We further extend IGF to MNIST and Fashion-MNIST, which contain images of different sizes than CIFAR. The reconstructed results, shown in Figure 13, reveal that the images are nearly indistinguishable from the originals based on gradient differences. This high-quality reconstruction is achieved through our composite optimization approach, which combines $\mathcal{L}_M$ with $\mathcal{L}_P$ loss. This combination ensures that the reconstructed images not only match the original images at the pixel level but also maintain perceptual similarity in terms of high-level features.

Table 2: Reconstruction performance across three metrics on five common defense mechanisms.

| Defense Method | None | Gradient Pruning | | | Sign Compression | Gauss Noise | Gradient Perturb | Gradient Smooth |
|---|---|---|---|---|---|---|---|---|
| | | 0.7 | 0.8 | 0.9 | 0.001 | 0.1 | 0.01 | 0.1 |
| MSE ↓ | 0.0211 | 0.0216 | 0.0221 | 0.0222 | 0.0225 | 0.0298 | 0.0197 | 0.0232 |
| PSNR ↑ | 17.19 | 17.0758 | 17.0694 | 17.0521 | 16.9704 | 15.7044 | 17.6116 | 16.8371 |
| LPIPS ↓ | 0.3261 | 0.3704 | 0.3796 | 0.3810 | 0.3796 | 0.4011 | 0.3663 | 0.3852 |

**Reconstruction Performance against Defense Mechanisms.**

We evaluate the reconstruction performance of IGF against five common defense mechanisms on *ConvNet* at the *sample-level* using EFU. The technical details of these defenses are provided in Appendix C.3, with results summarized in Table 2. Against Gradient Pruning (with hyperparameters $\{0.7, 0.8, 0.9\}$), Sign Compression, and Gradient Smoothing, IGF maintains comparable performance to the no-defense baseline, achieving MSE values around 0.022, PSNR around 17, and LPIPS around 0.38. Among the defenses, Gaussian Noise proves comparatively robust, while Gradient Perturbation is the comparatively weakest. Overall, IGF delivers reasonable reconstruction quality with negligible degradation relative to the no-defense setting. This robustness arises from our learning-based inversion model, which exhibits strong mapping capabilities. These findings underscore IGF's substantial resilience, enabling it to largely bypass existing defenses and recover forgotten data effectively. Consequently, they highlight the pressing need for novel defense strategies that can fundamentally impair an attacker's ability to reconstruct meaningful information.

**Reconstruction Performance against Orthogonal Obfuscation Defense.** As shown in the Figure 5, our proposed Orthogonal Obfuscation Defense disrupts reconstruction by altering gradient difference directions while preserving their L2-norm. Reconstructed images exhibit random noise, effectively thwarting IGF and protecting sensitive data. A detailed theoretical analysis of the Orthogonal Obfuscation Defense is provided in Appendix H.

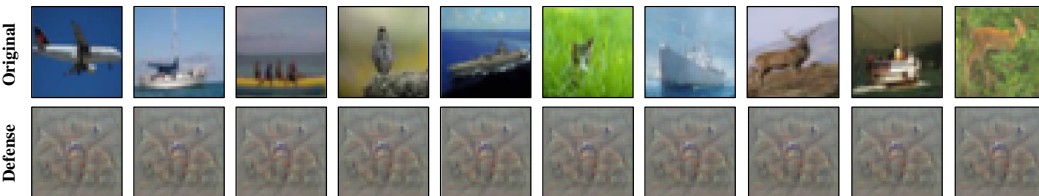

Figure 5: Forgotten images and our reconstructed images on the CIFAR-10 dataset under Orthogonal Obfuscation defense on *ConvNet* at the *sample-level* using EFU.

## 4.3 ABLATION STUDIES

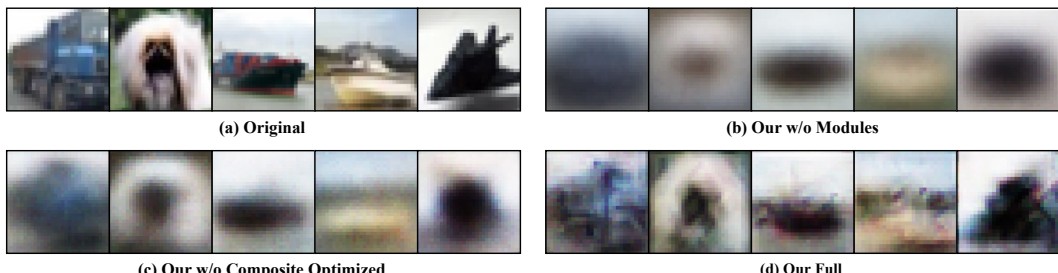

Figure 6: Forgotten images and our reconstructed images using inversion model across different component configurations on *ConvNet* at the *sample-level* using EFU.

To evaluate the effectiveness of our proposed composite loss optimization module and pixel-level inversion model in the attack framework, we conduct ablation studies to visualize the reconstruction results under various configurations, as shown in Figure 6. The **Original** row depicts the ground-truth forgotten samples. The **Our w/o Modules** variant, which utilizes MSE as the loss function alongside a simple three-layer multi-layer perceptron as the inversion model, yields severely degraded reconstructions characterized by pronounced artifacts and substantial loss of structural integrity. This outcome highlights the intrinsic difficulties of reconstruction attacks and underscores the indispensable value of our proposed enhancements. The **Our w/o Composite Optimized** configuration, which preserves the pixel-level inversion model but employs MSE for loss computation, generates images that maintain rudimentary shapes yet are plagued by blurring, chromatic aberrations, and deficient fine-grained details. This emphasizes the pivotal role of perceptual losses in distilling high-level semantic attributes that transcend basic pixel-wise fidelity. By contrast, our full model (**Our Full**), which integrates two main proposed components, achieves reconstructions with significantly improved visual quality. These images exhibit sharper definition, better texture preservation, and more accurate color reproduction. By effectively balancing low-level pixel information and high-level semantic features, our comprehensive approach yields reconstructions that closely resemble the original forgotten samples. **Further ablation studies on federated aggregation methods, auxiliary datasets, dimensionality reduction techniques, and the hyperparameter $\beta$ are provided in Appendix E.**

## 5 CONCLUSION

In this paper, we expose a critical privacy vulnerability in FU by proposing a novel reconstruction attack that exploits gradient differences used as PoFU. Our proposed IGF leverages the latent correlations between gradient differences and forgotten samples to reconstruct large-scale private data from individual PoFU. Through extensive experiments, we demonstrate that our attack achieves high-fidelity reconstruction, exposing the inadequacy of existing FU safeguards. To counter this threat, we introduce an orthogonal obfuscation defense that disrupts the reconstruction process, forcing inverted images into fixed noise patterns that resist reconstruction. Our findings underscore the fragility of current FU mechanisms against gradient-based and gradient-difference-based attacks, highlighting the urgent need for robust defenses and motivating further exploration of secure FU strategies.

## ETHICS STATEMENT

This research adheres to the ICLR Code of Ethics, which all authors have read and committed to follow during the submission process. The study involves analyzing potential privacy vulnerabilities in Federated Unlearning (FU) systems, specifically focusing on the reconstruction of forgotten data using gradient differences as Proof of Federated Unlearning (PoFU). While the work aims to enhance privacy protections by identifying and mitigating these vulnerabilities, it raises concerns regarding privacy and security issues.

The proposed Inverting Gradient difference to Forgotten data (IGF) attack framework and the orthogonal obfuscation defense mechanism were developed using publicly available benchmark datasets (e.g., CIFAR-10, MNIST) and do not involve human subjects or real-world personal data. However, the theoretical capability of reconstructing forgotten samples could have implications if applied to sensitive data, potentially leading to privacy breaches. To address this, we emphasize that our defense mechanism is designed to preserve PoFU verification utility while preventing such reconstructions, thereby supporting compliance with data privacy regulations like GDPR.

## REPRODUCIBILITY STATEMENT

We have made every effort to ensure the reproducibility of our work. The details of the model architecture, training process, and hyperparameters are provided in Section 4.1 and Appendix F. A complete description of the experimental setup, including datasets, models, and evaluation metrics, is included in Section 4.1 and Appendix C. Algorithmic details and proofs of theoretical claims are presented in Appendix H.

This study adheres to the principles of open science, emphasizing transparency and accessibility in research. The source code accompanying this work is publicly available on Anonymous GitHub at https://anonymous.4open.science/r/IGF. The repository provides artifact instructions, dependencies, core codes (*e.g.*, data, models, evaluation), and scripts, in compliance with ICLR's reproducibility policy.

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

# Verifiably Forgotten? Gradient Differences Still Enable Data Reconstruction in Federated Unlearning

## Table of Contents for Appendix.

## A    DISCUSSION AND LIMITATIONS

To the best of our knowledge, IGF framework is the first to exploit gradient differences as an attack surface in federated unlearning (FU). Previous reconstruction attacks in machine learning and federated learning (FL) (Zhu et al., 2019; Geiping et al., 2020; Sun et al., 2024; Wu et al., 2023) directly leverage *sample-level* gradients, which inherently contain richer sample information. Currently, the only available baseline for reconstruction attacks in FU is GIAMU (Hu et al., 2024), which relies on *white-box* access to both the original and unlearned models and overlooks the privacy vulnerabilities arising from gradient differences sharing during verification. Our work addresses this gap by demonstrating that an *honest-but-curious* adversary with partial prior knowledge can reconstruct forgotten samples by inverting gradient differences. Additionally, in extreme scenarios, such as a black-box setting where the adversary lacks prior knowledge or cannot exploit the directionality of gradient differences, the attack's complexity increases significantly, making the reconstruction of forgotten data largely unexplored.

## B    TECHNICAL REFERENCE MATERIALS

This section provides essential technical references to support the understanding and implementation of the proposed Inverting Gradient difference to Forgotten data (IGF) framework. Table 3 consolidates key notations and equations used throughout the paper, offering a concise reference for readers to interpret the methodology and results. Complementing this, the pseudocode (Algorithm 1) outlines the IGF framework's procedural steps, facilitating a clear and reproducible depiction of the attack mechanisms. Together, these materials enhance the accessibility of the technical content, helping to engage with the proposed methods efficiently.

Table 3: Mathematical notations

| Notation | Description | Notation | Description |
|---|---|---|---|
| $\mathcal{C}$ | Set of Clients | $\mathcal{D}$ | Global Dataset |
| $H$ | The Number of Clients | $\mathbf{M}$ | Original Global Model |
| $^u\mathbf{M}$ | Unlearned Global Model | $\mathbf{g}$ | Stochastic Gradient |
| $\mathcal{G}$ | Gradient Queried by Adversary | $\ell$ | Loss Function in Local Training |
| $T$ | Number of Global Rounds | $\mathcal{I}$ | Inversion Model |
| $(x, y)$ | Data Point | $B$ | Batch Size |
| $\phi(\cdot)$ | Intermediate Feature Extractor | $\Delta\mathcal{G}, \Delta\mathbf{g}$ | Gradient Differences |
| $d$ | Model Size | $\mathbf{U}$ | Left Singular Vectors |
| $\mathbf{V}$ | Right Singular Vectors | $\mathbf{r}$ | Random Vector |
| $\mathcal{M}$ | Mask Matrix in Gradient Pruning | $\epsilon$ | Gaussian Noise |
| $w$ | Window Size in Gradient Smoothing | $\zeta$ | Parameter Deviation Constraint Radius |
| $\mathbf{\Psi}$ | Gradient Differences Matrix | $\mathbf{V}^{[k]}$ | Projection Matrix |

## C    EXPERIMENTAL SETTINGS

### C.1    DATASETS

We evaluate our proposed IGF method on the widely adopted CIFAR-10 and CIFAR-100 datasets (Krizhevsky et al., 2009), which serve as standard benchmarks in the fields of reconstruction attacks and federated learning. CIFAR-10 consists of 60,000 color images ($32 \times 32$ pixels) spanning 10 categories, with 50,000 images allocated for training and 10,000 for testing, ensuring 6,000 images per category. Similarly, CIFAR-100 mirrors this structure but encompasses 100 categories, each containing 600 images, for a total of 60,000 images (50,000 training and 10,000 testing).

In addition, we assess our approach using the MNIST (LeCun et al., 1998) and Fashion-MNIST (Xiao et al., 2017) datasets. MNIST comprises 70,000 grayscale images ($28 \times 28$ pixels) of handwritten digits, split into 60,000 training and 10,000 testing images. Fashion-MNIST, conceived as a more demanding counterpart to MNIST, also includes 70,000 grayscale images ($28 \times 28$ pixels) across 10 categories of fashion items, adhering to the same training-testing division.

---

**Algorithm 1** Inverting Gradient difference to Forgotten data (IGF) framework

---

**Input:** Original model $\mathbf{M}$, unlearned model $^u\mathbf{M}$, auxiliary dataset $\mathcal{D}_{\text{aux}}$, PoFUs $\{\boldsymbol{\Delta}\mathbf{g}^{(n_i)}\}_{i \in \mathcal{C}_u}$, variance threshold $\nu$; Training parameters of the inversion model: loss trade-off $\beta$, learning rate $\eta$, epochs $E$.

**Output:** Reconstruction results $\{\hat{\mathbf{x}}^{(n_i)}\}_{i \in \mathcal{C}_u}$.

▷ **Inversion Model Training Phase**

1: **for** each $(x_i, y_i) \in \mathcal{D}_{\text{aux}}$ **do**                                 ▷ Preparation of Training Dataset
2:     Collects $\mathcal{G}_i, {}^u\mathcal{G}_i$ against Eq. (3),
3:     Computes $\boldsymbol{\Delta}\mathcal{G}_i = \mathcal{G}_i - {}^u\mathcal{G}_i$ and appends it into $\boldsymbol{\Delta}\mathcal{G}$
4: **end for**
5: Construct $\boldsymbol{\Psi} \leftarrow [\boldsymbol{\Delta}\mathcal{G}_1, \ldots, \boldsymbol{\Delta}\mathcal{G}_m]^\top$ where $m = |\mathcal{D}_{\text{aux}}|$     ▷ Gradient Differences Projection via SVD
6: Center $\boldsymbol{\Psi}^{\text{cen}} = \boldsymbol{\Psi} - \boldsymbol{\mu}\mathbf{1}_m^\top$ with mean vector $\boldsymbol{\mu} = \frac{1}{m}\sum \boldsymbol{\Delta}\mathcal{G}_i$
7: Perform SVD: $\mathbf{U}, \boldsymbol{\Sigma}, \mathbf{V}^\top \leftarrow \text{SVD}(\boldsymbol{\Psi}^{\text{cen}})$
8: Select $k$ via cumulative variance $\geq \nu$ as in Eq. (4)
9: Compute projected gradient differences $\boldsymbol{\Delta}\mathcal{G}^{\text{proj}} \leftarrow \boldsymbol{\Psi}\mathbf{V}^{[k]}$
10: Initialize inversion model $\mathcal{I}(\boldsymbol{\omega})$                                 ▷ Training Inversion Model
11: **for** epoch $= 1$ to $E$ **do**
12:     Optimize $\boldsymbol{\omega}$ by minimizing $\mathcal{L}_{\text{attack}} = \mathcal{L}_M + \beta\mathcal{L}_P$ as in Eq. (5)–(7)
13: **end for**
▷ **Privacy Exfiltration Phase**
14: **for** each $i \in \mathcal{C}_u, \mathcal{A}$ **do**
15:     Receive PoFU $\boldsymbol{\Delta}\mathbf{g}^{(n_i)}$
16:     Construct $\boldsymbol{\Psi}^{(n_i)}$ from $\boldsymbol{\Delta}\mathbf{g}^{(n_i)}$
17:     Project $\boldsymbol{\Delta}\mathbf{g}^{(n_i)\text{proj}} \leftarrow (\boldsymbol{\Psi}^{(n_i)} - \boldsymbol{\mu}\mathbf{1}_m^\top)\mathbf{V}^{[k]}$
18:     Reconstruct forgotten samples $\hat{\mathbf{x}}^{(n_i)}$ as in Eq. (8)
19: **end for**
20: **return** $\{\hat{\mathbf{x}}^{(n_i)}\}_{i \in \mathcal{C}_u}$

---

To investigate the influence of the auxiliary dataset's distribution relative to the forgetting dataset on IGF's reconstruction efficacy, we further employ the SVHN (Netzer et al., 2011) dataset. SVHN is a real-world image dataset designed for machine learning applications, particularly digit recognition, comprising over 600,000 labeled color images of house numbers sourced from Google Street View. These images, typically in $32 \times 32$ RGB format, feature digits from 0 to 9 and are available in single-digit or multi-digit sequence formats.

### C.2 DETAILS OF METRICS

MSE measures the average squared difference between the original forgotten image and the reconstructed image. It is widely used as a loss function in image processing tasks and image quality assessment $\text{MSE} = \frac{1}{N}\sum_{i=1}^{N}(x_i - \hat{x}_i)^2$, where $x_i$ is the pixel value of the original forgotten image and $\hat{x}_i$ is the pixel value of the reconstruction.

PSNR measures the quality of the reconstructed or compressed image relative to the forgotten image. It is expressed in decibels (dB) and is inversely related to MSE—lower MSE values correspond to higher PSNR values. $\text{PSNR} = 10 \cdot \log_{10}\left(\frac{R^2}{\text{MSE}}\right)$, Where $R$ is the maximum pixel value.

LPIPS (Zhang et al., 2018) is a perceptual similarity metric designed to assess the perceptual quality of images based on learned features from a neural network (typically a pretrained deep network like VGG). Unlike MSE and PSNR, LPIPS is more aligned with human visual perception, focusing on perceptual similarity rather than pixel-level accuracy $\text{LPIPS}(x, \hat{x}) = \frac{1}{L}\sum_{l=1}^{L}\|\phi_l(x) - \phi_l(\hat{x})\|_2^2$, where $L$ is the total number of layers used for feature extraction. $\|\cdot\|_2$ is the Euclidean distance (L2 norm) between the feature maps.

## C.3 DETAILS OF COMMON DEFENSE MECHANISMS

This section outlines five defense mechanisms (Wu et al., 2023) designed to obfuscate shared gradients and mitigate gradient-based reconstruction attacks through various perturbation techniques. Given an input gradient vector $\mathbf{g}$, each mechanism produces an obfuscated gradient vector $\mathbf{g}'$. We adapt these mechanisms to perturb shared gradient differences in FU.

(a) **Sign Compression.** The sign compression mechanism applies the sign operation to each component of the gradient $\mathbf{g}$, retaining only its sign ($-1$, $0$, or $1$) and discarding magnitude information. This preserves the gradient's direction while significantly reducing communication overhead, as only sign bits are transmitted. By limiting the attacker's access to sign information, this method increases the difficulty of reconstructing forgotten data. The operation is defined as:

$$\mathbf{g}' = \text{sign}(\mathbf{g}), \quad \text{where} \quad \text{sign}(\mathbf{g}_i) = \begin{cases} 1, & \text{if } \mathbf{g}_i > 0 \\ -1, & \text{if } \mathbf{g}_i < 0 \\ 0, & \text{if } \mathbf{g}_i = 0 \end{cases} \tag{10}$$

(b) **Gradient Pruning.** Gradient pruning sparsifies the gradient by retaining only the $k$ components with the largest absolute values, setting all others to zero. A binary mask $\mathcal{M}$ selectively preserves these significant components. Widely used in FL to reduce communication costs, this method also enhances privacy by limiting the attacker's access to a subset of gradient components, complicating the inference of forgotten data. The operation is formulated as:

$$\mathbf{g}' = \mathbf{g} \odot \mathcal{M}, \tag{11}$$

where $\odot$ denotes element-wise multiplication, and $\mathcal{M}$ is the mask matrix.

(c) **Gaussian Noise.** This mechanism perturbs the gradient $\mathbf{g}$ by adding independent and identically distributed Gaussian noise $\epsilon \sim \mathcal{N}(0, \sigma^2 \mathbf{I})$. Controlled by the standard deviation $\sigma$, the noise introduces uncertainty to achieve differential privacy, obscuring precise gradient values and hindering reconstruction of forgotten data. The operation is expressed as:

$$\mathbf{g}' = \mathbf{g} + \epsilon, \quad \epsilon \sim \mathcal{N}(0, \sigma^2 \mathbf{I}). \tag{12}$$

(d) **Gradient Perturbation.** This method perturbs the gradient by adding noise proportional to the gradient's magnitude, applying larger perturbations to dimensions with greater gradient values. The perturbed gradient is defined as:

$$\mathbf{g}' = \mathbf{g} + (\mathcal{N}(\mathbf{0}, \mathbf{I}) \times \text{scale}) \times (|\mathbf{g}| \times \text{factor}), \tag{13}$$

where $\mathcal{N}(\mathbf{0}, \mathbf{I})$ is a standard normal random tensor, scale determines the base perturbation magnitude, and factor adjusts the sensitivity of the perturbation to the gradient's amplitude.

(e) **Gradient Smoothing.** Gradient smoothing mitigates high-frequency variations in the gradient by applying a moving average over the feature dimensions, blending the result with the original gradient. The operation is formulated as:

$$\mathbf{g}' = \text{reshape}\left((1 - \alpha_{\text{gs}})\mathbf{g}^{\text{flat}} + \alpha_{\text{gs}} \cdot \text{MA}_w(\mathbf{g}^{\text{flat}})\right), \tag{14}$$

where $\mathbf{g}^{\text{flat}}$ is the flattened gradient, $\text{MA}_w$ denotes the moving average with window size $w$, and $\alpha_{\text{gs}} \in [0, 1]$ controls the smoothing intensity.

## D MORE COMPREHENSIVE BASELINE COMPARISON

### D.1 COMPARISON WITH LEARNING-BASED SOTA

The learning-based SOTA method, Learning To Invert (LTI) (Wu et al., 2023), leverages gradients from an auxiliary dataset as inputs for training the inversion model, incorporates hash-based dimensionality reduction, and employs mean squared error (MSE) as the loss function. Our proposed IGF surpasses LTI in several key aspects:

**Auxiliary Sample Efficiency.** Under the same CIFAR-10 dataset, IGF requires $\sim 10,000$ auxiliary samples to train the inversion model, whereas LTI demands $\sim 50,000$ samples, demonstrating superior efficiency in data utilization.

**Loss Function Innovation.** Our composite loss function explicitly optimizes both pixel-level accuracy and high-level semantic fidelity. Extensive ablation studies validate its pivotal role in enhancing reconstruction quality (Figure 6(d)), which is far beyond simple MSE loss used in prior works (Wu et al., 2023; Hu et al., 2024) (Figure 6(c)).

**Input Reduction for Inversion Model.** The gradient difference derived from large-scale global models like ResNet-20 exhibits $269,722$ dimensions. This high-dimensional output serves as input for inversion models, significantly increasing computational complexity. As detailed in Section E.5, our SVD-based approach outperforms hash-based dimensionality reduction (Wu et al., 2023) in both dimensionality reduction efficiency (Table 10) and numerical reconstruction accuracy (Figure 10).

## D.2 COMPARISON WITH OPTIMIZATION-BASED SOTA

Notably, few baseline attacks are directly comparable for FU reconstruction, as most existing methods either: i) Process gradients from a single model, or ii) Compare differences between two models. To enable fair comparison, we adapt input formats and optimization procedures of DLG (Zhu et al., 2019) to serve as baseline. The adapted algorithm is named DLGD (Deep Leakage from Gradient Difference) and presented in Algorithm 2.

---

**Algorithm 2** Deep Leakage from Gradient Difference (DLGD).

**Input:** Gradient difference $\Delta\mathbf{g}$; **Colluding client** who can compute the gradients on original model $\mathbf{M}(\cdot;\boldsymbol{\theta}^T)$ and unlearned model ${}^u\mathbf{M}(\cdot;\boldsymbol{\theta}')$
**Output:** private forgotten data $\mathbf{x}, \mathbf{y}$
The *honest-but-curious* auditor $\mathcal{A}$ executes:

1: $\mathbf{x}'_1 \leftarrow \mathcal{N}(0,1), \mathbf{y}'_1 \leftarrow \mathcal{N}(0,1)$          ▷ Initialize dummy inputs and labels.
2: **for** $i \leftarrow 1$ to $N_{iter}$ **do**
3:      Query $\nabla_{\boldsymbol{\theta}}\ell\big(\mathbf{M}(\mathbf{x}'_i;\boldsymbol{\theta}^T),\mathbf{y}'_i\big)$ and $\nabla_{\boldsymbol{\theta}}\ell\big({}^u\mathbf{M}(\mathbf{x}'_i;\boldsymbol{\theta}'),\mathbf{y}'_i\big)$ from **colluding client**.
4:      $\Delta\mathbf{g}'_i \leftarrow \nabla_{\boldsymbol{\theta}}\ell\big(\mathbf{M}(\mathbf{x}'_i;\boldsymbol{\theta}^T),\mathbf{y}'_i\big) - \nabla_{\boldsymbol{\theta}}\ell\big({}^u\mathbf{M}(\mathbf{x}'_i;\boldsymbol{\theta}'),\mathbf{y}'_i\big)$    ▷ Compute dummy gradient difference.
5:      $\mathbb{D}_i \leftarrow ||\Delta\mathbf{g}'_i - \Delta\mathbf{g}||^2$
6:      $\mathbf{x}'_{i+1} \leftarrow \mathbf{x}'_i - \eta\nabla_{\mathbf{x}'_i}\mathbb{D}_i, \mathbf{y}'_{i+1} \leftarrow \mathbf{y}'_i - \eta\nabla_{\mathbf{y}'_i}\mathbb{D}_i$    ▷ Update data to match gradient difference.
7: **end for**
8: **return** $\mathbf{x}'_{n+1}, \mathbf{y}'_{n+1}$

---

Our experiments incorporating optimization-based SOTA (DLGD) demonstrate that **IGF achieves lower communication overhead and superior reconstruction performance**.

**Communication Complexity.** As illustrated in Table 4, IGF incurs a fixed number of gradient differences from offline queries on the auxiliary dataset and zero online queries for reconstructing any number of forgotten samples in FU. Conversely, DLGD eliminates offline query overhead but scales linearly with the number of samples under a fixed iteration count $N_{\text{iter}}$.

Table 4: Comparison of offline and online computational complexity.

| Method | Offline | Online |
|---|---|---|
| Optimization-based SOTA (DLGD) | 0 | $\sim 2 \times N_{\text{iter}} \times \theta \times N_u$ |
| Our proposed IGF | $\sim 2 \times N_{\text{aux}} \times \theta$ | 0 |

$N_{\text{aux}} \approx$ thousands, $N_{\text{iter}} \approx$ hundreds, $\theta =$ model size and $N_u$ is the number of forgotten samples to be reconstructed.

**Superior Reconstruction Accuracy.** We evaluate DLGD's reconstruction performance across varying iteration counts ($N_{\text{iter}} = 100, 200, 300, 500$) in Table 5. The optimal numerical reconstruction is observed at $N_{\text{iter}} = 200$. Nonetheless, IGF consistently outperforms DLGD in reconstruction quality across all evaluated $N_{\text{iter}}$ values.

Furthermore, visualizations of DLGD's reconstructed samples in Figure 7 illustrate its inability to extract meaningful signals from gradient differences, resulting in visually degraded outputs resembling

Table 5: Reconstruction performance of IGF vs. DLGD with different iterations $N_{\text{iter}}$.

| Metric | IGF | DLGD(100) | DLGD(200) | DLGD(300) | DLGD(500) |
|---|---|---|---|---|---|
| MSE ↓ | **0.0211** | 0.5378 | 0.4494 | 0.7022 | 1.6185 |
| PSNR ↑ | **17.19** | 2.6935 | 3.4730 | 1.5349 | -2.0914 |
| LPIPS ↓ | **0.3261** | 0.4142 | 0.3855 | 0.4232 | 0.4779 |

random noise. This limitation stems from the challenges in optimizing randomly generated dummy gradient differences to recover forgotten samples, aligning with observations in Ju et al. (2025). Thus, directly applying optimization-based methods to gradient differences yields suboptimal reconstruction performance.

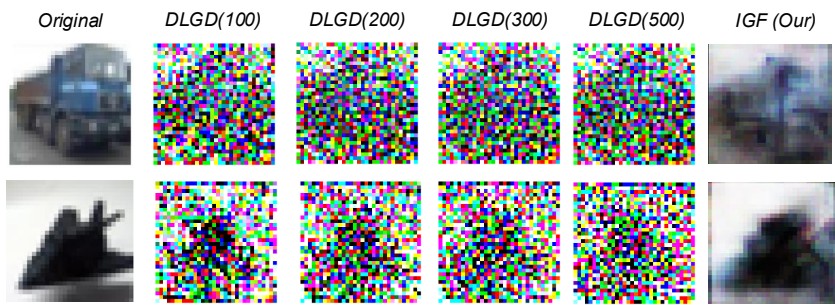

Figure 7: Reconstruction images of IGF vs. DLGD with different iterations $N_{\text{iter}}$.

**This observation aligns with our perspective: gradient differences for the same sample, compared to gradients, contain limited and mixed signals, making reconstruction more challenging.**

**Efficient Large-Scale Reconstruction.** We further quantify runtime overhead (in seconds) for reconstructing varying numbers of forgotten samples ($N_u$), with $N_{\text{iter}} = 200$. Table 6 highlights IGF's scalability, amortizing its fixed 3,020-second training overhead across samples. In contrast, DLGD's per-sample optimization leads to linear runtime growth with $N_u$.

Table 6: Reconstruction runtime (in seconds) of IGF and DLGD for varying forgotten samples $N_u$, assuming $N_{\text{iter}} = 200$.

| | $N_u = 0$ | $N_u = 50$ | $N_u = 200$ | $N_u = 400$ | $N_u = 600$ | $N_u = 800$ |
|---|---|---|---|---|---|---|
| IGF | 3020 | 3045 | 3220 | 3320 | 3420 | 3520 |
| DLGD | 0 | 3300 | 13200 | 26400 | 39600 | 52800 |

# E   ADDITIONAL ABLATION STUDIES

## E.1   RECONSTRUCTION PERFORMANCE AGAINST DEFENSE MECHANISMS ON *ResNet20*

To further validate the robustness and generalizability of IGF, we conducted additional ablation studies evaluating its performance against common defense mechanisms using *ResNet20* as the global model backbone. As shown in Table 7, we assessed the IGF attack against five common defense mechanisms. Without defenses, our attack achieves MSE=0.04, PSNR=13.51, and LPIPS=0.43 on CIFAR-10. With defenses applied, the attack remains highly effective, with MSE ranging from 0.04 to 0.06, PSNR from 12.06 to 13.51, and LPIPS from 0.43 to 0.54, indicating that these defenses fail to disrupt our attack significantly. This aligns with our conclusion that current defense mechanisms that disrupt our attack significantly are inadequate against gradient-difference-based reconstruction attacks, underscoring the need for our proposed orthogonal obfuscation defense.

Table 7: Reconstruction performance across three metrics at five common defense mechanisms on *ResNet20*.

| Defense Method | None | Gradient Pruning | | | Sign Compression | Gauss Noise | Gradient Perturb | Gradient Smooth |
|---|---|---|---|---|---|---|---|---|
| | | 0.7 | 0.8 | 0.9 | 0.001 | 0.1 | 0.01 | 0.1 |
| MSE $\downarrow$ | 0.0445 | 0.0585 | 0.0584 | 0.0585 | 0.0445 | 0.0607 | 0.0585 | 0.0581 |
| PSNR $\uparrow$ | 14.05 | 12.3307 | 12.33 | 12.3254 | 13.5144 | 12.0673 | 12.3270 | 12.3275 |
| LPIPS $\downarrow$ | 0.4607 | 0.5272 | 0.5316 | 0.5364 | 0.4389 | 0.5452 | 0.5345 | 0.5357 |

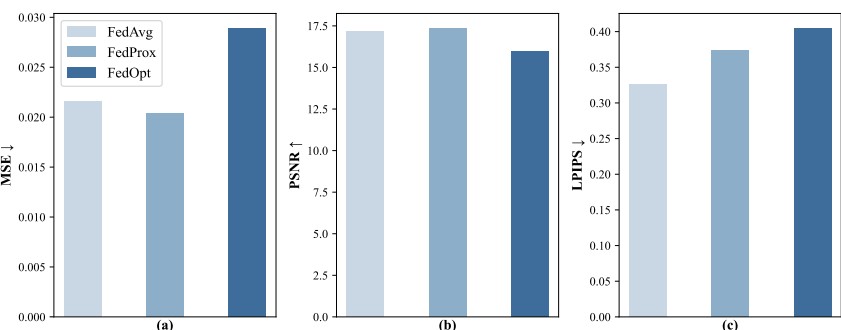

Figure 8: The reconstruction performance under different federated aggregation methods.

## E.2 IMPACT OF DIFFERENT FEDERATED AGGREGATION METHODS

We investigated how three federated aggregation methods, including FedAvg (McMahan et al., 2017), FedProx (Li et al., 2020), and FedOpt (Reddi et al., 2020), affect the performance of reconstruction attacks in FU scenarios. Figure 8 illustrates the performance of our attack method across various aggregation algorithms commonly used in FL systems. The results demonstrate that while aggregation methods can influence reconstruction quality, our attack remains effective across different techniques. When examining more sophisticated aggregation methods like FedProx and FedOpt, we observe slightly different reconstruction patterns, but the overall attack effectiveness remains consistent.

## E.3 IMPACT OF DIFFERENT DISTRIBUTIONS OF AUXILIARY DATASETS

We select SVHN (Netzer et al., 2011) as the auxiliary dataset for out-of-distribution (OOD) tasks. Compared to CIFAR-10, SVHN exhibits entirely distinct visual features and semantic categories, rendering it a dataset with a markedly different distribution. To quantify the degree of OOD in the auxiliary dataset, we introduce the variable $\alpha$. The auxiliary dataset consists of $10,000$ samples, with SVHN comprising a proportion $\alpha$ and CIFAR-10 comprising $1 - \alpha$. Specifically, when $\alpha = 0$, the auxiliary dataset is in-distribution with the forgotten dataset, while $\alpha = 1$ corresponds a fully OOD auxiliary dataset.

Table 8: The results of *ConvNet* as the FL model at different degrees of OOD.

| $\alpha$ | 0.0 | 0.1 | 0.3 | 0.5 | 0.7 | 0.9 | 1.0 |
|---|---|---|---|---|---|---|---|
| MSE $\downarrow$ | 0.0211 | 0.02066 | 0.0219 | 0.0236 | 0.0261 | 0.0306 | 0.0395 |
| PSNR $\uparrow$ | 17.1947 | 17.3447 | 17.1221 | 16.8155 | 16.449 | 15.829 | 14.837 |
| LPIPS $\downarrow$ | 0.3261 | 0.373 | 0.387 | 0.408 | 0.436 | 0.489 | 0.55 |

Table 9: The results of *ResNet20* as the FL model at different degrees of OOD.

| $\alpha$ | 0.0 | 0.1 | 0.3 | 0.5 | 0.7 | 0.9 | 1.0 |
|---|---|---|---|---|---|---|---|
| MSE ↓ | 0.0445 | 0.0598 | 0.05991 | 0.05997 | 0.06034 | 0.0615 | 0.06404 |
| PSNR ↑ | 14.05 | 12.7890 | 12.7867 | 12.7682 | 12.762 | 12.6687 | 12.50563 |
| LPIPS ↓ | 0.4607 | 0.5199 | 0.52849 | 0.53568 | 0.54463 | 0.5456 | 0.57495 |

We present the numerical results of *ConvNet* and *ResNet20* serving as FL model at different degrees of OOD in Table 8 and 9, respectively. Based on the above experimental results, we can draw the following conclusions: Higher OOD degrees ($\alpha$ approaching 1) result in relatively poorer reconstruction performance, as the auxiliary data becomes semantically and categorically unrelated to the forgotten dataset. When the auxiliary dataset includes a proportion of in-distribution data (e.g., $\alpha = 0.1, 0.3, 0.5$), the reconstruction quality of our proposed IGF method remains high.

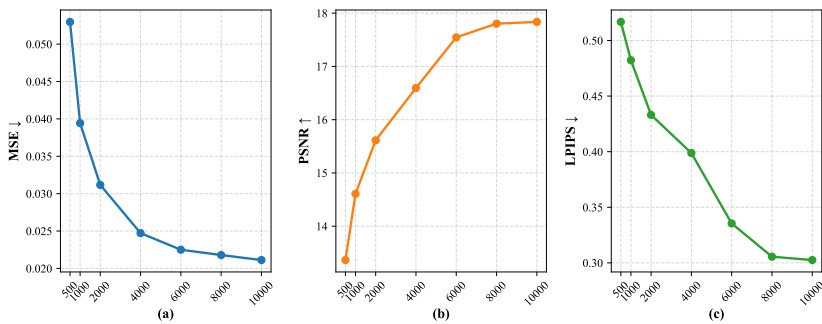

Figure 9: The reconstruction performance with different auxiliary dataset sizes.

### E.4 IMPACT OF DIFFERENT AUXILIARY DATASET SIZES

We investigate the influence of varying auxiliary dataset sizes on the efficacy of our attack method. As illustrated in Figure 9, we incrementally scale the dataset from 500 to 10,000 samples. Experimental results reveal that performance metrics stabilize when the auxiliary dataset comprises approximately 8,000 to 10,000 samples, demonstrating that our method achieves efficient and robust performance without requiring extensive auxiliary data. Notably, even with a modest dataset size, our proposed attack method effectively leverages available knowledge to deliver high-quality image reconstruction.

### E.5 COMPARATIVE ABLATION STUDY OF DIMENSIONALITY REDUCTION METHODS

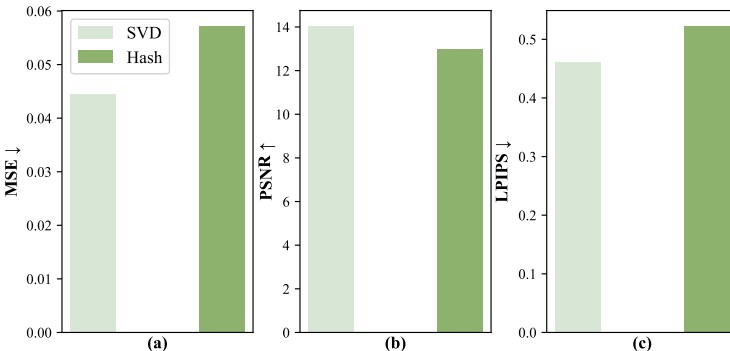

Figure 10: Comparison of the reconstruction effectiveness with applying SVD and Hash dimensionality reduction.

To gain a deeper understanding of the effectiveness of dimensionality reduction methods, we compared the performance of Hash-based dimensionality reduction and Singular Value Decomposition (SVD) in terms of reduction quality and reconstruction results. Hash-based dimensionality reduction (Weinberger et al., 2009) is a vector compression method that relies on random projection, mapping high-dimensional gradient differences to a lower-dimensional space through a sparse random matrix. Specifically, a sparse matrix is constructed where each high-dimensional vector component is randomly assigned to a lower-dimensional target dimension, and each reduced dimension represents the cumulative sum of the corresponding high-dimensional gradient differences. This approach is computationally efficient and well-suited for rapidly compressing gradient differences. However, its randomness disregards the inherent structure of the gradient differences, potentially leading to significant information loss.

As shown in Figure 10, SVD outperforms the reconstruction after Hash dimensionality reduction in both reconstruction effects, and as shown in Table 10 achieves more significant dimensionality reduction by extracting only key information. SVD-based dimensionality reduction is a data-driven method that decomposes the covariance matrix of the gradient differences to extract principal component directions as the projection basis. SVD dynamically selects the number of dimensions to retain a substantial portion of the variance (e.g., 95%), ensuring that the reduced results capture the primary patterns of the original gradient differences.

| Method | Size |
|---|---|
| Original | 269722 |
| Hash | 134861 |
| SVD | 433 |

Table 10: Comparison of the effectiveness of SVD and Hash for gradient differences reduction.

SVD outperforms Hash-based reduction because it prioritizes the retention of critical information while minimizing the impact of irrelevant noise. Furthermore, in reconstruction tasks, SVD-preserved gradient differences maintain structured features, enabling inversion models to more effectively learn the mapping from lower-dimensional features to the original data, resulting in higher-quality reconstructed images. Conversely, Hash-based reduction disrupts the gradient differences structure through random mixing, making it challenging for reconstruction networks to disentangle useful information, which often leads to blurry or distorted reconstructed images.

### E.6 COMPARATIVE ABLATION STUDY OF DIFFERENT $\beta$ PARAMETERS

To investigate the role of the parameter $\beta$ in the loss function, which governs the trade-off between pixel-level accuracy and perceptual quality, we conduct an ablation study to assess its impact on reconstruction attack performance. Specifically, we evaluate the effect of varying $\beta \in \{0.1, 1.0, 2.0\}$ on three key metrics: MSE, PSNR, and LPIPS. As shown in Figure 11, increasing $\beta$ reveals a clear trade-off: pixel-level accuracy degrades, as indicated by worsening MSE, and perceptual quality diminishes, as reflected by deteriorating LPIPS, while PSNR exhibits a peak at an intermediate $\beta$ before declining. These findings underscore $\beta$'s critical role in mediating the balance between pixel-wise fidelity and high-level perceptual features.

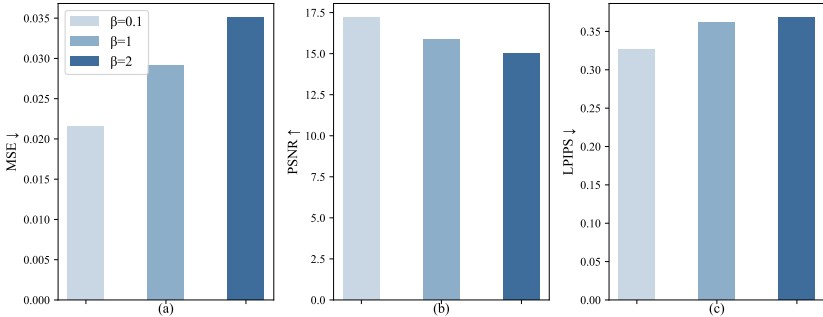

Figure 11: The reconstruction performance under different $\beta$.

# F    INVERSION MODEL ARCHITECTURE

As illustrated in Figure 12, our pixel-level inversion model features a carefully designed architecture comprising multiple Conv2d and BatchNorm2d layers. We incorporate PixelShuffle for effective upsampling, minimizing artifacts in reconstructed results. A linear layer paired with an initial Reshape operation enhances input processing, while a final Sigmoid activation and Reshape ensure high-quality output generation.

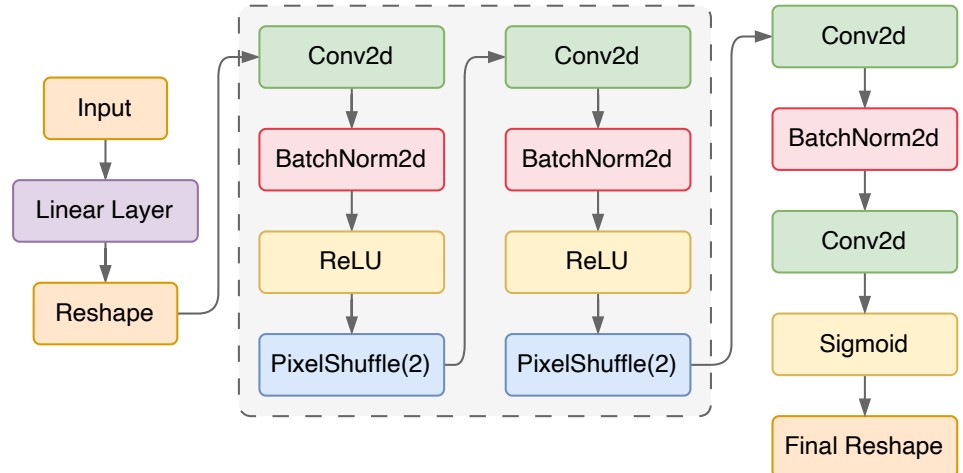

Figure 12: Architecture of the proposed pixel-level inversion model.

# G    ADDITIONAL RECONSTRUCTED IMAGES

This section showcases the forgotten images and their corresponding reconstructions across multiple datasets, as presented in Figures 13, 14, and 15. In each figure, **odd columns display the original images**, and **even columns show our reconstructed results**. Specifically, Figure 13 illustrates the reconstruction results of IGF using *ConvNet* in the *sample-level* scenario on the MNIST and Fashion-MNIST datasets. Due to the lower complexity of the MNIST series images, gradient information can more effectively capture key features, resulting in reconstructed images that closely resemble the originals. In contrast, Figure 14 presents the reconstruction results of IGF using *ConvNet* in the *sample-level* scenario on the CIFAR-100 dataset. Despite the complex distribution of the CIFAR-100 dataset, IGF can still generate reconstructed images with a certain level of effectiveness. For the scenario of *class-level* unlearning, Figure 15 presents the forgotten images and reconstruction results on CIFAR-10 for the unlearned class (car).

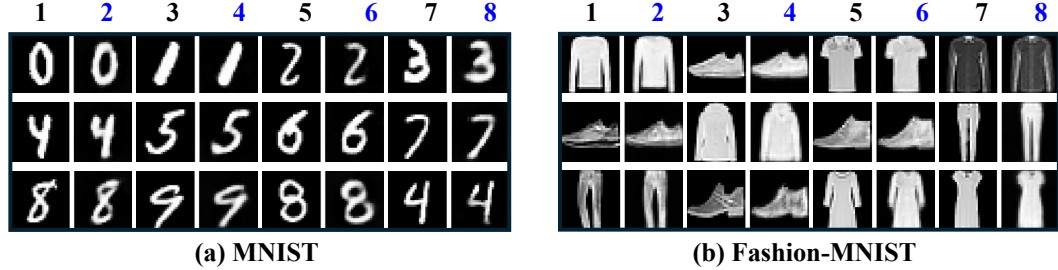

(a) MNIST          (b) Fashion-MNIST

Figure 13: Forgotten and reconstructed images on MNIST and Fashion-MNIST within 1,000 randomly forgotten samples.

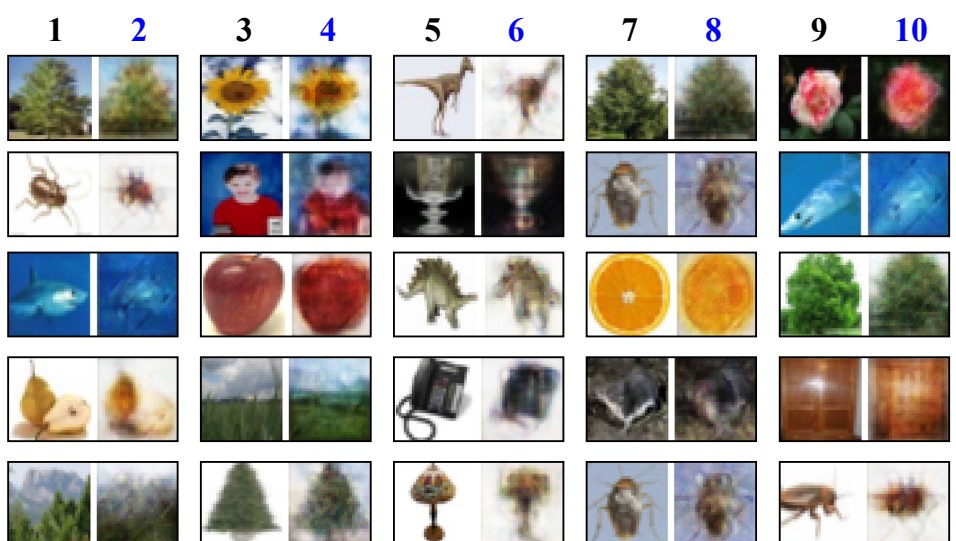

Figure 14: Forgotten and reconstructed images on CIFAR-100.

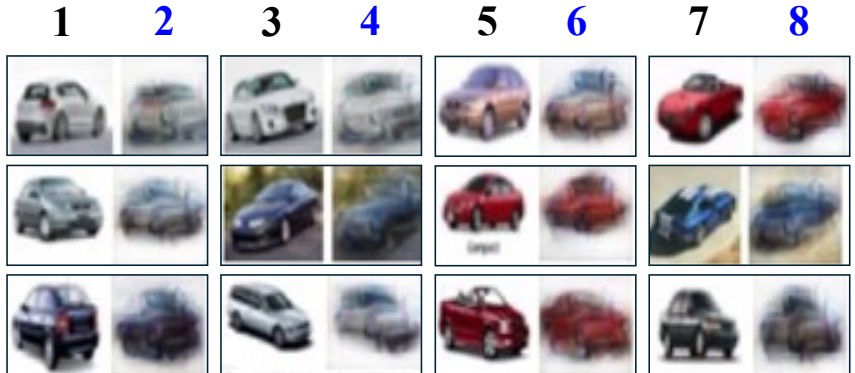

Figure 15: Forgotten and reconstructed images on CIFAR-10 for the unlearned class (car).

## H    THEORETICAL ANALYSIS OF ORTHOGONAL OBFUSCATION DEFENSE

We aim to construct a mapping $f : \Delta g \mapsto \Delta g^{\text{obf}}$ such that:

$$(\Delta g^{\text{obf}})^\top \Delta g = 0, \qquad \|\Delta g^{\text{obf}}\|_2 = \|\Delta g\|_2. \tag{15}$$

Let the original gradient difference vector be $\Delta g \in \mathbb{R}^d$, and let $r \sim \mathcal{N}(0, I_d)$ be a random Gaussian vector. Define

$$u = r - \frac{r^\top \Delta g}{\|\Delta g\|_2^2} \Delta g, \qquad \Delta g^{\text{obf}} = \|\Delta g\|_2 \cdot \frac{u}{\|u\|_2}. \tag{16}$$

**Orthogonality.**

$$(\Delta g^{\text{obf}})^\top \Delta g = \left( \|\Delta g\|_2 \cdot \frac{u}{\|u\|_2} \right)^\top \Delta g = \frac{\|\Delta g\|_2}{\|u\|_2} \cdot (u^\top \Delta g). \tag{17}$$

Since

$$u^\top \Delta g = \left( r - \frac{r^\top \Delta g}{\|\Delta g\|_2^2} \Delta g \right)^\top \Delta g = r^\top \Delta g - \frac{r^\top \Delta g}{\|\Delta g\|_2^2} \cdot \Delta g^\top \Delta g = r^\top \Delta g - r^\top \Delta g = 0, \tag{18}$$

we obtain

$$(\Delta g^{\text{obf}})^\top \Delta g = 0. \tag{19}$$

**Norm Preservation.**

$$\|\Delta g^{\text{obf}}\|_2 = \left\| \|\Delta g\|_2 \cdot \frac{u}{\|u\|_2} \right\|_2 = \|\Delta g\|_2 \cdot \left\| \frac{u}{\|u\|_2} \right\|_2 = \|\Delta g\|_2 \cdot 1 = \|\Delta g\|_2. \tag{20}$$

If $u = 0$, then

$$r = \frac{r^\top \Delta g}{\|\Delta g\|_2^2} \Delta g \iff r \in \text{span}(\Delta g). \tag{21}$$

Since $r \sim \mathcal{N}(0, I_d)$ is drawn from a continuous distribution, the probability of $r$ lying in the one-dimensional subspace spanned by $\Delta g$ is zero for $d > 1$. Hence this case can be ignored in practice.

$$\boxed{\Delta g^{\text{obf}} \text{ is orthogonal to } \Delta g \text{ and has the same norm.}}$$

The construction is based on Gram-Schmidt orthogonalization. A random Gaussian vector $r$ is chosen, and its projection onto $\Delta g$ is subtracted to yield $u$, which is guaranteed to be orthogonal to $\Delta g$. To ensure that the obfuscated vector preserves the same magnitude as the original one, $u$ is normalized to a unit vector and scaled by $\|\Delta g\|_2$. This guarantees both orthogonality and norm preservation. The only degenerate case occurs if $r$ is collinear with $\Delta g$, but this event has probability zero in continuous distributions and is negligible in practice.

## I    THE USE OF LARGE LANGUAGE MODELS (LLMS)

In the preparation of this paper, a large language model (LLM) was used solely for minor text polishing and grammar corrections. The LLM did not contribute to research ideation, content generation, or any other significant aspect of the work. All content, including the final text, has been thoroughly reviewed and approved by the authors, who take full responsibility for its accuracy and originality.

