# OpenReview forum: "Verifiably Forgotten? Gradient Differences Still Enable Data Reconstruction in Federated Unlearning"
_ICLR.cc/2026/Conference — ICLR 2026 Conference Withdrawn Submission_

### Official Review · Reviewer_5xMG · 2025-10-22

**Soundness:** 1
**Presentation:** 2
**Contribution:** 1
**Rating:** 2
**Confidence:** 4

**Summary:**

The paper investigates privacy vulnerabilities in Federated Unlearning (FU), a setting where clients can “forget” specific data while maintaining global model utility. The authors demonstrate that Proof of Federated Unlearning (PoFU)—often expressed as gradient differences between original and unlearned models—can leak sensitive information. They propose Inverting Gradient difference to Forgotten data (IGF), an attack that reconstructs forgotten samples using singular value decomposition (SVD) for dimensionality reduction and a deep inversion model with a composite pixel–perceptual loss.

**Strengths:**

1. In FU, auditing is an important research question, as well as its privacy risks.
2. The authors conducted broad experiments to present the performance of the proposed attack.

**Weaknesses:**

1. The scenario lacks solid evidence. The authors assume that the PoFU will share the gradient difference with the auditor, which lacks sufficient discussion on related works, in order to present that such an auditing method is general or widely used.
2. Besides, why does a sufficiently small L2 norm of gradient difference indicate successful forgetting? The authors should explain more about it, because this appears unreasonable (the gradient difference should be sufficiently large rather than small), and it lacks reliable guarantees for auditing purposes.
3. The assumption is too strong. The authors assume that the auditor could query the original/unlearned models with auxiliary data, which seems impractical in an FL setting, since the auditor has little reason to possess such high-level privileges.
4. There is a very similar work in the field of gradient leakage attacks [a], where an honest-but-curious server could utilize the gradients to reconstruct the private training data via a trained network. I understand that the authors utilize the gradient difference as input, but it seems that the two papers share a very similar idea.

[a] Wu R, Chen X, Guo C, et al. Learning to invert: Simple adaptive attacks for gradient inversion in federated learning[C]//Uncertainty in Artificial Intelligence. PMLR, 2023: 2293-2303.

**Questions:**

1. Why does a sufficiently small L2 norm of gradient difference indicate successful forgetting? And, is it a general approach?
2. How/Why could an auditor query the original/unlearned models with auxiliary data in practice?
3. What is the difference between the work and [a]? It seems that the proposed method in [a] could be easily applied to this scenario.

---

### Official Review · Reviewer_Q8of · 2025-10-27

**Soundness:** 2
**Presentation:** 3
**Contribution:** 2
**Rating:** 4
**Confidence:** 3

**Summary:**

The paper “Verifiably Forgotten? Gradient Differences Still Enable Data Reconstruction in Federated Unlearning” investigates the privacy risks in federated unlearning (FU) systems, particularly those employing proof-of-federated-unlearning (PoFU) protocols. The authors propose a new inversion-based attack framework, IGF (Inverting Gradient difference to Forgotten data), showing that gradient differences, used to verify unlearning, can inadvertently leak information about the forgotten samples. The IGF framework combines SVD-based dimensionality reduction with a learned inversion model to reconstruct data from gradient differences. The paper also introduces an Orthogonal Obfuscation defense, which preserves the L2 norm of PoFU gradients for verification purposes while disturbing their directional information to prevent data leakage.
Extensive experiments on CIFAR-10/100, MNIST, and other datasets demonstrate that IGF can reconstruct forgotten samples with high fidelity, and that Orthogonal Obfuscation can effectively mitigate this risk.

**Strengths:**

1. Novel Problem Formulation: The paper identifies a previously underexplored vulnerability in federated unlearning verification, namely, that even verifiable unlearning proofs may unintentionally reveal private information. This perspective bridges two research domains, machine unlearning and gradient inversion, making the work conceptually novel and timely.
2. Methodological Creativity: IGF cleverly integrates dimensionality reduction (SVD) and learned gradient inversion, significantly reducing computational cost compared to optimization-based inversion methods. The proposed Orthogonal Obfuscation defense is intuitive yet effective, offering a practical mitigation that retains verification utility.
3. Technical Quality and Experimental Depth: The experiments are extensive, covering multiple datasets, models, and ablation studies. The implementation of both attack and defense is well-documented and reproducible.
4. Significance: The study provides valuable insight into the privacy-verifiability trade-off in federated unlearning systems. Even if the attack scenario is somewhat idealized, the findings highlight an important design tension in real-world FU protocols, which could influence future privacy-preserving verification designs.

**Weaknesses:**

While the paper is technically competent and raises a novel concern about the privacy of FU verification, several key assumptions and modeling choices significantly limit its practical relevance and generality.

1. Overly Strong Assumption on Auxiliary Data (D_aux):
The attack assumes that the auditor possesses an auxiliary dataset drawn from the same or nearly identical distribution as the forgotten data. This is a strong and often unrealistic assumption in real-world federated scenarios, where data distributions across clients are highly heterogeneous and access to similar data by auditors is unlikely. The paper mentions out-of-distribution (OOD) data only briefly in the appendix but does not quantify how performance degrades as distributional shift increases.
Moreover, when unlearning aims to remove specific individual samples rather than entire concepts or classes, auxiliary data provides little benefit because it lacks instance-level correspondence. The IGF attack, therefore, is only meaningful for concept-level unlearning, not for itemized sample deletion.

2. Limited Realism of Gradient Query Access:
The gray-box assumption—that an auditor can query gradients from both the original model (M) and the unlearned model (uM)—is questionable in practical PoFU deployments. Most FU auditing interfaces expose only scalar verification signals (e.g., norms or hashes), not per-sample gradients. Although this assumption may be used for upper-bound analysis, it should be clearly delineated as a non-standard access condition, not as a general property of PoFU systems.

3. Unrealistic Per-Sample PoFU Setting:
The attack depends on PoFU submissions being per-sample, i.e., each forgotten sample contributes its individual gradient difference. However, practical FU systems usually aggregate or anonymize such information for privacy and efficiency. If the PoFU were aggregated (summed or averaged across multiple samples), the directional signal of individual gradients would be lost, making the attack ineffective. The authors do not evaluate this more realistic setting.

4. Unproven Directionality Intuition:
The key design intuition, IGF relies heavily on the directionality of gradient differences because they encode input image features, is plausible but only empirically motivated. No theoretical justification (e.g., via gradient–input mutual information, cosine similarity, or feature-space correlation analysis) is provided. This intuition underpins both the attack and defense; thus, more rigorous analysis is necessary to substantiate it.

**Questions:**

To make the paper more convincing and broadly relevant, I suggest the following clarifications and additions:

1. Clarify the Auxiliary Data Requirement:
Please quantify how IGF’s performance changes as the auxiliary dataset becomes more out-of-distribution. How much auxiliary data is needed for meaningful inversion? If the unlearning target is sample-specific (rather than class-level), can IGF still succeed?

2. Clarify Gradient Query Access Assumption:
While I understand this assumption as an upper-bound analysis, please clearly describe under what concrete PoFU implementations such gradient access is possible. This will help readers assess the practical risk boundary.

3. Clarify PoFU Granularity and Its Impact:
Confirm whether IGF’s success depends on per-sample PoFU submission. If so, please explicitly state this in the main text and discuss how aggregation (sum/average) would affect reconstruction. Optionally, a small-scale experiment comparing per-sample vs. aggregated PoFU would strengthen the paper.

4. Validate the Directionality Hypothesis:
Provide quantitative evidence that gradient difference directions correlate with input feature spaces (e.g., through cosine similarity between ∆g and feature gradients, or an information-theoretic measure). This will substantiate the core intuition behind both the attack and the proposed defense.

5. Broader Discussion on Practical Implications:
Given that many of these assumptions (same-distribution auxiliary data, per-sample PoFU, gradient access) are strong, the authors might consider repositioning the contribution as a theoretical upper bound on potential leakage rather than as a direct real-world threat. A clear boundary statement would make the paper both honest and valuable.

---

### Official Review · Reviewer_eDk8 · 2025-10-29

**Soundness:** 2
**Presentation:** 3
**Contribution:** 2
**Rating:** 2
**Confidence:** 4

**Summary:**

This paper identifies a critical privacy vulnerability in Federated Unlearning, demonstrating that the very Proof of Federated Unlearning, specifically, the gradient differences sent to an auditor for verification, can be exploited to reconstruct the supposedly forgotten private data. The authors introduce the IGF attack to prove this reconstruction is feasible and, as a solution, propose a novel orthogonal obfuscation defense designed to protect the proof's privacy while maintaining its verifiability.

**Strengths:**

+ The problem statement and motivation are clear.
+ The paper is generally well written.
+ The method and the proposed defense are discussed in reasonable detail.

**Weaknesses:**

The paper's central threat model relies on an honest-but-curious auditor receiving a PoFU. This premise is not sufficiently justified. Why can't the client who requested the unlearning simply act as their own auditor by downloading the unlearned global model and verifying the data's removal directly (e.g., via membership inference)? What specific limitations of such a direct, client-side verification justify the introduction of a third-party auditor?

Considering the paper's premise that a third-party auditor is required, the entire attack vector still relies on an unnecessarily weak verification model that exposes raw gradient differences. The authors should address why a standard, privacy-preserving cryptographic method (such as a zero-knowledge proof [1]) is not considered. Such an approach would allow the server to prove the unlearning computation was performed correctly without revealing any sensitive information, thereby neutralizing the proposed attack.

The reliance on a "representative" auxiliary dataset to train the inversion model is a strong, unproven assumption. This is highly impractical for the most critical use cases, such as unlearning unique personal photos, proprietary documents, or medical scans, where no such representative dataset would be publicly available for an auditor.

The attack's success is likely highly sensitive to the unlearning batch size. In practical scenarios, unlearning requests would be batched, and the aggregated gradients from a large, diverse batch would be far less informative. Could the authors provide an ablation study on batch size to show how IGF performs as k increases?

The IGF attack is presented as an empirical method without any formal guarantees. The paper would be significantly strengthened by a theoretical analysis of the inversion model, including its convergence behavior, computational complexity, and the specific identifiability conditions that determine when reconstruction from gradient differences is mathematically guaranteed to succeed or fail.

The authors must position their work in relation to a relevant paper, DRAUN [2]. This paper tackles the same data reconstruction problem in federated unlearning, notably claiming to be algorithm-agnostic.

The paper's generalizability is limited, as the attack is only demonstrated on small-scale models and low-resolution datasets. The authors provide no evidence that this attack is feasible on modern, large-scale architectures, such as ViT, or can reconstruct complex, high-resolution data from ImageNet.


[1] T. Eisenhofer et al., "Verifiable and Provably Secure Machine Unlearning". SATML 2025.

[2] H. Lamri et al., "DRAUN: An Algorithm-Agnostic Data Reconstruction Attack on Federated Unlearning Systems". arXiv 2025.

**Questions:**

1. What specific limitation of direct client-side verification justifies the necessity of a third-party auditor?
2. How practical is the assumption that an auditor possesses a representative auxiliary dataset for truly private or unique data?
3. Can the authors provide an ablation study on the unlearning batch size (k) to demonstrate how reconstruction fidelity degrades as k increases?
4. Can the authors provide evidence that this attack is feasible on large-scale models like ViT and high-resolution datasets like ImageNet?

---

### Official Review · Reviewer_dV1o · 2025-10-30

**Soundness:** 2
**Presentation:** 2
**Contribution:** 3
**Rating:** 4
**Confidence:** 4

**Summary:**

This paper investigates privacy vulnerabilities in Federated Unlearning (FU), revealing that using gradient differences as Proof of Federated Unlearning (PoFU) allows adversaries to reconstruct forgotten data. The authors propose IGF (Inverting Gradient difference to Forgotten data), a learning-based reconstruction attack leveraging Singular Value Decomposition (SVD) for dimensionality reduction and a pixel-level inversion model with a composite loss. Additionally, they introduce an Orthogonal Obfuscation Defense that preserves verification integrity while perturbing gradient directions to mitigate reconstruction risks.

**Strengths:**

1. Novel viewpoint – The first to reveal privacy leakage via gradient differences in PoFU.

2. Thorough experimentation – Covers diverse datasets, architectures, and FU scenarios.

3. Strong writing and organization – Clear structure and academic rigor suitable for ICLR publication.

**Weaknesses:**

1. Strong assumptions – The auditor’s access to gradients and auxiliary datasets is unrealistic in many FU deployments (e.g., DP, sparsification).

2. Limited defense evaluation – The defense lacks quantitative utility–privacy trade-off metrics.

3. Narrow interpretation of results – The analysis focuses mainly on reconstruction fidelity, not systemic implications for FU protocols.

4. Experiments are limited to image datasets; generalization to NLP or tabular FU tasks remains untested.

5. The robustness under out-of-distribution auxiliary datasets is not thoroughly evaluated.

6 . The scalability of the defense in large, heterogeneous federated environments is uncertain.

**Questions:**

see weakness

---

### Note · Authors · 2025-11-13

I have read and agree with the venue's withdrawal policy on behalf of myself and my co-authors.